# Discovery of the most compact 3+1-type quadruple star system TIC 120362137

**Tamás Borkovits** [1,2,3,22] ✉, **Saul A. Rappaport** [2,4,22], **Hai-Liang Chen** [5,6,22], **Guillermo Torres** [7,22], **Tibor Mitnyan** [2,8], **Veselin B. Kostov** [9], **Brian P. Powell** [9], **Theo Pribulla** [10], **Petr Zasche** [11], **Imre B. Bíró** [1,2], **István Csányi** [1], **Donát R. Czavalinga** [2,8], **Zoltán Dencs** [3,12], **Zoltán Garai** [10,12], **Jakub Kolář** [13], **Pavel Cagaš** [14,15,16], **Zbyněk Henzl** [16,17], **Tom Kaye** [18], **Hana Kučáková** [11,16,19,20], **Martin Mašek** [16,21] & **Robert Uhlař** [16]

Hierarchical multiple stellar systems with short outer periods comprise an important subgroup of multiple star systems. In this paper we present the discovery and spectro-photodynamical analysis of the most compact known 3+1 quadruple stellar system, TIC 120362137. Through investigations of the observations made with the TESS satellite and ground-based follow up measurements, we find that the system consists of an eclipsing binary with a few-day-period that in turn eclipses, and is eclipsed by, a third star on a $P_{mid}$ = 51.3 d orbit. This inner subsystem, which contains three stars that are more massive and hotter than the Sun, is more spatially compact than Mercury's orbit around our Sun, and is orbited by a fourth Sun-like star with a period $P_{out}$ = 1046 d. We detect the spectral lines of all four stars, making this system the most thoroughly studied 3+1 type quadruple stellar system. The future evolution of TIC 120362137 is also modeled, and we conclude that this entire system will likely end up as a pair of white dwarfs.

Hierarchical multiple stellar systems, especially those in which even the orbital periods of the wider subsystem(s) range only from months to a few years, comprise a small but important subgroup of the triple and multiple star zoo. In regard to triple stellar systems, their hierarchical nature, which is required for the long-term dynamical stability of such systems, means that two of the three stars comprise an inner binary, while the third component revolves on a much wider orbit around the center of mass of the inner binary. Hence, the motion of the three stars can be described approximately as a composite of two (perturbed) Keplerian motions. Then, regarding a hierarchical

[1]Baja Astronomical Observatory, University of Szeged, Baja, Hungary. [2]HUN–REN-SZTE Stellar Astrophysics Research Group, Baja, Hungary. [3]Konkoly Observatory, HUN–REN Research Centre for Astronomy and Earth Sciences, Budapest, Hungary. [4]Department of Physics, Kavli Institute for Astrophysics and Space Research, M.I.T., Cambridge, MA, USA. [5]Yunnan Observatories, Chinese Academy of Sciences (CAS), Kunming, PR China. [6]International Centre of Supernovae, Yunnan Key Laboratory, Kunming, PR China. [7]Center for Astrophysics, Harvard & Smithsonian, Cambridge, MA, USA. [8]Department of Experimental Physics, Institute of Physics, University of Szeged, Szeged, Hungary. [9]NASA Goddard Space Flight Center, Greenbelt, MD, USA. [10]Astronomical Institue, Slovak Academy of Sciences, Tatranská Lomnica, Slovakia. [11]Astronomical Institute, Faculty of Mathematics and Physics, Charles University, Praha 8, Czechia. [12]Gothard Astrophysical Observatory, ELTE Eötvös Loránd University, Szombathely, Hungary. [13]Department of Theoretical Physics and Astrophysics, Faculty of Science, Masaryk University, Brno, Czechia. [14]Institute of Theoretical Physics, Faculty of Mathematics and Physics, Charles University, Praha 8, Czechia. [15]BSObservatory, Zlín, Czechia. [16]Variable Star and Exoplanet Section, Czech Astronomical Society, Ondřejov, Czechia. [17]Hvězdárna Jaroslava Trnky ve Slaném, Slaný 1, Czechia. [18]Raemor Vista Observatory, Foundation for Scientific Advancement, Sierra Vista, AZ, USA. [19]Astronomical Institute, Academy of Sciences, Ondřejov, Czechia. [20]Research Centre for Theoretical Physics and Astrophysics, Institute of Physics, Silesian University in Opava, Opava, Czechia. [21]FZU—Institute of Physics of the Czech Academy of Sciences, Prague, Czechia. [22]These authors contributed equally: Tamás Borkovits, Saul A. Rappaport, Hai-Liang Chen, Guillermo Torres. ✉e-mail: borko@bajaobs.hu

quadruple star system, it may have two substantially different configurations. In the case of a hierarchical 2 + 2 type quadruple, there are two inner binaries, while the centers of mass of these binaries revolve around each other with a much wider separation, i.e., a third Keplerian orbit. In contrast to this, the 3 + 1 or (2 + 1) + 1 type systems are composed of a hierarchical triple star subsystem, as described above, while the fourth star revolves around the center of mass of the inner triple on an orbit which has a much larger semi-major axis. According to Tokovinin[1], it is very likely that these two types of quadruple stellar systems are formed in different ways.

Such compact multiple star systems are important astrophysical objects because they (i) often exhibit short timescale (order of the period of the triple's orbit) dynamical interactions that can lead to a determination of most of the stellar and orbital parameters of the system[2–5]; (ii) can provide insight into how the basic processes of star formation can produce such systems; (iii) can shed some light on how some close binaries may form[6–9]; and (iv) may lead to the formation of some exotic objects containing collapsed stars[10–14].

A close, compact multiple star system (or a circumbinary planet) can be detected and studied most easily if it contains at least one eclipsing binary (EB). Several such multiples have been discovered in the last decade due to the observations of the exoplanet-hunter space telescopes *Kepler*[15] and Transiting Exoplanet Survey Satellite (TESS)[16]. Many of them have displayed rare or never-before-seen effects during the intervals of the highly precise observations by these spacecraft. These are, for example, extra or outer eclipse events (where the members of the inner, close binary subsystem occult and/or are occulted by a third stellar or planetary companion—see Carter et al.[17] and Doyle et al.[18], for the first detections of a triply eclipsing triple star and a transiting circumbinary planet, respectively); rapid eclipse depth variations (sometimes even the disappearance of the stellar eclipses[19]); rapid, uneven, and retrograde apsidal motion[20]; unusually large, sometimes spike-shaped eclipse timing variations (ETV)[19,20]; and last, but not least, the appearance of two or, in exceptional cases, three EBs in a single system[21].

What is common among these systems, as well as in some multiply transiting exoplanetary systems, is the need for new analysis methods for the precision satellite and ground-based observations of these objects. Besides several new phenomena which have now been routinely observed (for example, multiple and extra eclipses and/or transits), the presence of gravitational perturbations leads to the continuous departure of the orbital motions and, hence, the instantaneous positions and velocities of each body from its pure, unperturbed Keplerian motion. These measurable perturbations can even occur on timescales of just weeks or months. This makes it imperative to integrate numerically the motions of the bodies simultaneously with the light curve, ETV, and spectroscopic analyses of the observations. Such models and analyses are called "(spectro-)photodynamical" and were first used in connection with *Kepler*-spacecraft observations of KOI 126, the first known triply eclipsing triple star by Carter et al.[17], and Kepler-16 the first circumbinary planetary system by Doyle et al.[18]. Since then, photodynamical analysis has been in widespread use when analyzing and modeling multiply eclipsing stellar systems, as well as transiting exoplanetary systems.

Starting in the year 2019, due to the more than 6-year-long monitoring campaign by the TESS satellite, several formerly unknown and record-setting multiple stellar systems have been identified. Regarding triple stars, Kostov et al.[22] reported the discovery of the most compact triple star system, TIC 290061484. The record holder 2 + 2 type quadruple star system (at least at the end of 2025), BU CMi, which consists of two eccentric EBs, was also found by TESS[23], and the number of such kinds of objects is also growing continuously[24]. In contrast to these, only a very few 3 + 1-type quadruple star systems are known.

Here we show the identification and detailed spectro-photodynamical analysis of TIC 120362137. We find that, at the time of writing this paper, this system is the most compact known 3 + 1 type quadruple stellar system. We identify the signals of all four stars in the composite spectra and present radial velocity (RV) data for all the stellar components. Combining these RV data with the photometric light curves obtained through space-borne satellite observations, together with ground-based follow-up measurements as well as ETV data, we provide a consistent spectro-photodynamical solution for the system. The results include numerous dynamical and astrophysical parameters of the four constituent stars. We find that three of the four stars are more massive than our Sun, and could all fit within an area comparable to Mercury's orbit around our Sun. Moreover, the fourth, more distant component is similar to our Sun and orbits closer to the other three stars than the distance of Jupiter from our Sun. Finally, we carry out evolutionary simulations which make it very likely that this system of four stars will finish its life as a binary system of two white dwarf stars, which originated from the merger of the primordial stars.

## Results
### Discovering and revealing the quadruple nature of TIC 120362137
TESS observed TIC 120362137 during nine sectors between 2019 and 2024. These TESS light curves show the clear signal of a $P_{in}$ = 3.28 day-period EB. Moreover, nine 1–2 day-long extra fadings were detected during the nine sectors, which reveal that a third star also revolves around the center of mass of the EB with a period of $P_{mid}$ = 51.3 d. (In the two panels of Fig. 1, two different illustrative characteristic extra eclipse events are shown, while all nine observed third-body eclipses are plotted in Supplementary Fig. S3). Besides the extra eclipses, the ETV curve (see Fig. 2), which was determined from the mid-eclipse times of the EB (and tabulated in Supplementary Table S3), also exhibited the 51.3-day-periodic cycles, confirming the presence of the third star.

The brightness of the system made it possible to organize ground-based photometric and spectroscopic follow-up campaigns. The details of the instruments and the data reduction methods during these observing campaigns are described later in the Methods subsection Space-borne and ground-based observations. While the photometric follow-up observations were intended to obtain several new ETV points through observations of the regular eclipses of the innermost EB (though parts of some new third-body eclipses were also observed—see Supplementary Fig. S4), the spectroscopic measurements were primarily aimed at getting precise RV data for each star in the system.

These follow-up observations, together with the newer TESS data sets, clearly revealed that the occurrence times (and also the shapes and the amplitudes) of the third-body eclipses can be well modeled only by introducing some extra non-linearities into the motions of the three stars. Moreover, an extra non-linear and most likely cyclic variation in the mid-eclipse times of the regular eclipses of the innermost EB was also found. Therefore, we concluded, with considerable confidence, that there must be a fourth, more distant stellar component in this system. Hence, we made efforts to identify this fourth component directly in the spectroscopic data.

In order to find the RV signals of all four stars and calculate individual RVs for all components, from the best quality Tillinghast Reflector Echelle Spectrograph (TRES)[25,26] spectroscopic data (see in Methods subsection Space-borne and ground-based observations), we used the software package QUADCOR[27]. This is a four-dimensional extension of the software package TODCOR, which applies the two-dimensional cross-correlation technique[28], and which is in widespread use. In such a manner, besides the evident, readily visible lines of the two inner components (stars Aa and Ab), as well as the strongly

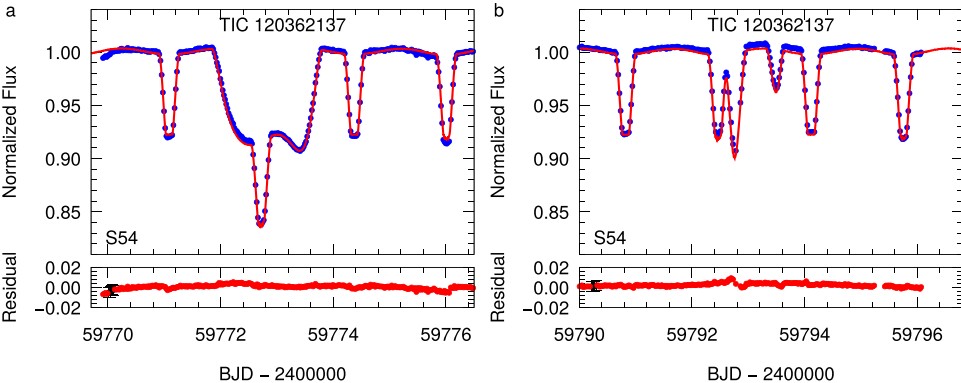

**Fig. 1 | Two typical third-body eclipses of TIC 120362137 from TESS Sector 54.**
**a** A third-body eclipse which had occurred around a primary eclipse of the inner binary and, therefore, the three stars were seen almost along a straight line by TESS; hence, the two inner stars eclipsed the more distant tertiary star at the same time, producing only a single long-duration extra eclipse (superposed on the regular primary eclipse of the inner binary). **b** During the next event, the tertiary star eclipsed the two inner binary members separately; hence, one could detect two extra dips (the first of which had begun before the end of a regular primary eclipse of the inner pair). Note also that the sharp, regular eclipses are the usual primary and secondary eclipses of the inner eclipsing binary. It can readily be seen that every second regular eclipse has a flat bottom, indicating total (secondary) eclipses. Blue points represent the observed, but 1800-s-binned TESS data, while the red smooth curves represent the best-fit photodynamical solutions (see Methods subsection Spectro-photodynamical analysis). The lower parts of both panels display the residual light curves. At the beginning of both residual curves, a few typical observational error bars are shown in black. Source data are provided as a Source data file.

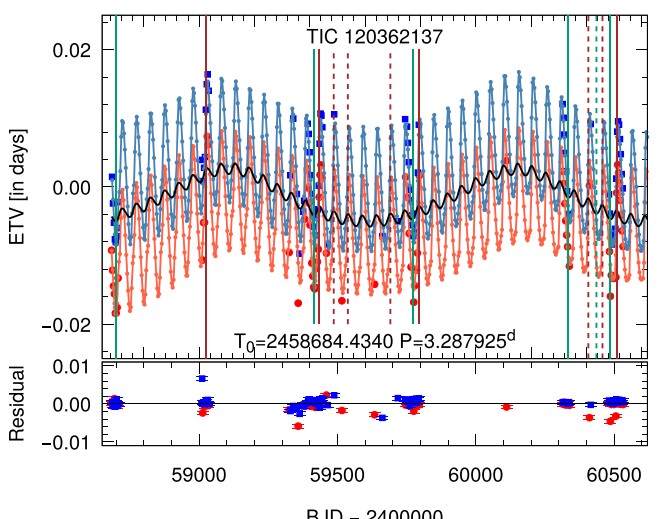

**Fig. 2 | Primary and secondary ETV curves of TIC 120362137.** The large red circles and blue boxes denote the primary and secondary ETV points derived from the TESS and ground-based follow-up observations, while the small red and blue dots, connected with similarly colored straight lines, stand for the primary and secondary ETV curves of the best-fit photodynamical solution (see Methods subsection Spectro-photodynamical analysis). The horizontally centered black curve represents the pure LTTE contribution. Vertical lines mark the times of the observed outer eclipses (green and brown -- the binary occulting the tertiary star and vice versa, respectively). Note, also, that the solid vertical lines refer to those third-body eclipses which were observed with TESS, while the dashed ones denote such events where some portions were observed during our ground-based photometric follow-up campaign. In the lower part of the figure, the residual points are shown, together with the uncertainties in the times of each individual eclipsing minimum. (The corresponding error bars, in general, are smaller than the sizes of the symbols) Source data are provided as a Source data file.

rotationally broadened signal of the third star B, it was also possible to detect the sharp but very faint lines of the fourth component, C. We were thereby able to determine the RV values individually for all four stars. The RVs and their uncertainties for all four components of TIC 120362137 are listed in Supplementary Table S4. Cross-correlation plots for two of the spectra are plotted in Fig. 3. The final $T_{\mathrm{eff}}$ values used for the templates for stars Aa, Ab, B, and C are 6750, 6750, 7000, and 5750 K, respectively. We also determined the fractional light contributions at the mean wavelength of our observations (about 5187 Å) using QUADCOR, and obtained $0.547 \pm 0.044$, $0.143 \pm 0.021$, $0.290 \pm 0.056$, and $0.020 \pm 0.010$, in the same order as above. We note, finally, that the measured projected rotational velocities of stars Aa and Ab (i.e., $v \sin i$ values of 49 and 26 km s$^{-1}$ for the two stars, respectively) are very close to the values that would be expected for synchronous rotation in the 3.28-day inner orbit (see Results subsection Current physical and dynamical parameters).

## Current physical and dynamical parameters

We carried out a joint spectro-photodynamical analysis (see "Methods" subsection Spectro-photodynamical analysis for the technical details) on all the available observational material of the $3+1$-type quadruple system TIC 120362137. The results are summarized in Tables 1 and 2. For each fitted parameter, the median values of the posteriors are given from the Markov-chain Monte Carlo (MCMC) analysis, together with the lower and upper $1-\sigma$ statistical uncertainties. In the same tables, we also give several derived (and not directly adjusted) additional parameters. Examples of the simultaneously fitted light curve solutions overplotted on the observational data are shown in the two panels of Fig. 1, while the measured ETV and RV points are shown plotted against the corresponding model fits in Figs. 2 and 4, respectively. Moreover, the architecture and the true physical size of the system are illustrated on the two panels of Fig. 5.

According to the presented results, three of the four stars of the system are clearly more massive and hotter than our Sun. The most massive component is the primary of the innermost binary (star Aa) with $M_{\mathrm{Aa}} = 1.75\,M_{\odot}$, $R_{\mathrm{Aa}} = 3\,M_{\odot}$, and $T_{\mathrm{Aa}} = 6609$ K, and slightly evolved from the main sequence (MS). The secondary star in the EB has $M_{\mathrm{Ab}} = 1.36\,M_{\odot}$, $R_{\mathrm{Ab}} = 1.5\,M_{\odot}$, and is actually somewhat hotter at $T_{\mathrm{Ab}} = 6725$ K. These properties clearly explain the shapes and relative depths of the regular eclipses of the innermost EB. The similar depths of the primary and secondary eclipses arise from the similar surface brightnesses (i.e., similar $T_{\mathrm{eff}}$), while the long flat bottoms of every other inner eclipse indicate the very different sizes of the two stars. It is interesting to note that the statistical uncertainties which are around 1% (or, in a few cases, well below this; see Tables 1–2). This precision is mainly due to the acquisition of accurate RV data for all the stellar components.

**Table 1 | Orbital and other dynamical parameters of TIC 120362137 from the joint photodynamical lightcurve, ETV, RV, SED, and `PARSEC` isochrone solution**

| orbital elements | | | |
|---|---|---|---|
| | **subsystem** | | |
| | **Aa–Ab** | **A–B** | **AB–C** |
| $t_0$ [BJD - 2400000] | 58683.3 | | |
| $P$ [days] | $3.284187^{+0.000049}_{-0.000052}$ | $51.3111^{+0.0046}_{-0.0039}$ | $1045.5^{+7.3}_{-8.4}$ |
| $a$ [$R_\odot$] | $13.576^{+0.037}_{-0.046}$ | $96.62^{+0.32}_{-0.34}$ | $769.9^{+4.4}_{-4.3}$ |
| $e$ | $0.00735^{+0.00017}_{-0.00019}$ | $0.22367^{+0.00071}_{-0.00073}$ | $0.274^{+0.013}_{-0.013}$ |
| $\omega$ [deg] | $199.4^{+1.5}_{-1.5}$ | $190.05^{+0.34}_{-0.30}$ | $289.0^{+4.7}_{-4.9}$ |
| $i$ [deg] | $88.70^{+0.17}_{-0.18}$ | $89.05^{+0.09}_{-0.09}$ | $90.3^{+8.9}_{-6.0}$ |
| $\mathcal{T}_0^{\mathrm{inf/sup}}$ [BJD - 2400000] | $58684.4268^{+0.0003}_{-0.0003}$ | $58701.8027^{+0.0100}_{-0.0110}$ | |
| $\tau$ [BJD - 2400000] | $58683.790^{+0.014}_{-0.013}$ | $58661.077^{+0.045}_{-0.040}$ | $59147^{+11}_{-11}$ |
| $\Omega$ [deg] | 0.0 | $0.26^{+0.21}_{-0.17}$ | $0.10^{+1.39}_{-1.34}$ |
| $(i_{\mathrm{mut}})_{A-...}$ [deg] | – | $0.47^{+0.26}_{-0.25}$ | $5.7^{+5.0}_{-3.9}$ |
| $(i_{\mathrm{mut}})_{B-...}$ [deg] | $0.47^{+0.26}_{-0.25}$ | – | $5.6^{+4.7}_{-3.7}$ |
| $\varpi^{\mathrm{dyn}}$ [deg] | $19.4^{+1.5}_{-1.5}$ | $10.08^{+0.33}_{-0.30}$ | $108.9^{+4.7}_{-4.9}$ |
| $i^{\mathrm{dyn}}$ [deg] | $3.9^{+3.5}_{-2.6}$ | $3.8^{+3.1}_{-2.4}$ | $1.9^{+1.5}_{-1.2}$ |
| $\Omega^{\mathrm{dyn}}$ [deg] | $8^{+23}_{-20}$ | $0^{+22}_{-21}$ | $182^{+31}_{-27}$ |
| $i_{\mathrm{inv}}$ [deg] | $89.9^{+6.0}_{-4.0}$ | | |
| $\Omega_{\mathrm{inv}}$ [deg] | $0.14^{+0.94}_{-0.90}$ | | |
| mass ratio [$q = M_{\mathrm{sec}}/M_{\mathrm{pri}}$] | $0.779^{+0.003}_{-0.003}$ | $0.478^{+0.005}_{-0.005}$ | $0.220^{+0.010}_{-0.009}$ |
| $K_{\mathrm{pri}}$ [km s$^{-1}$] | $91.56^{+0.36}_{-0.36}$ | $31.58^{+0.30}_{-0.27}$ | $6.93^{+0.27}_{-0.25}$ |
| $K_{\mathrm{sec}}$ [km s$^{-1}$] | $117.58^{+0.41}_{-0.46}$ | $66.17^{+0.25}_{-0.28}$ | $31.59^{+0.35}_{-0.38}$ |
| $V_\gamma$ [km s$^{-1}$] | $-22.12^{+0.09}_{-0.08}$ | | |
| Apsidal and nodal motion-related parameters | | | |
| $P_{\mathrm{apse}}$ [year] | $6.56^{+0.06}_{-0.06}$ | $34.79^{+0.08}_{-0.07}$ | $896^{+33}_{-24}$ |
| $P_{\mathrm{apse}}^{\mathrm{dyn}}$ [year] | $3.33^{+0.02}_{-0.02}$ | $5.66^{+0.02}_{-0.02}$ | $203.3^{+6.1}_{-8.6}$ |
| $P_{\mathrm{node}}^{\mathrm{dyn}}$ [year] | $6.73^{+0.04}_{-0.04}$ | | $265.8^{+6.6}_{-5.0}$ |
| $\Delta\omega_{\mathrm{3b}}$ [arc-sec/cycle] | $3115^{+19}_{-19}$ | $32187^{+133}_{-132}$ | $18271^{+721}_{-548}$ |
| $\Delta\omega_{\mathrm{GR}}$ [arc-sec/cycle] | $1.894^{+0.010}_{-0.013}$ | $0.414^{+0.003}_{-0.003}$ | $0.0650^{+0.0007}_{-0.0006}$ |
| $\Delta\omega_{\mathrm{tide}}$ [arc-sec/cycle] | $388^{+12}_{-12}$ | $1.167^{+0.036}_{-0.036}$ | $0.0188^{+0.0007}_{-0.0007}$ |

The meaning of each parameter[a] is as follows: $t_0$ is the epoch time for the osculating orbital elements; $P$ is the orbital period; $a$ the semimajor axis; $e$ the eccentricity; $\omega$ the argument of pericenter (of the secondary)[b]; $\varpi$ the longitude of periastron (of the secondary); $i$ the inclination; $\mathcal{T}_0^{\mathrm{inf/sup}}$ the time of conjunction of the secondary[c]; $\tau$ the time of periastron passage; $\Omega$ the longitude of the node relative to the node of the innermost orbit; $i_{\mathrm{mut}}$ the mutual inclination angle; $i_{\mathrm{inv}}$ the inclination of the system's invariable plane; $\Omega_{\mathrm{inv}}$ the longitude of the node of the invariable plane; $q$ the mass ratio (secondary/primary); $K_{\mathrm{pri/sec}}$ the RV amplitudes of the primary/secondary stars; $V_\gamma$ the systemic RV of the entire quadruple system; $P_{\mathrm{apse}}$ the nominal, theoretical apsidal motion period, $P_{\mathrm{node}}$ the nominal, theoretical nodal regression period; while $\Delta\omega_{\mathrm{3b,GR,tide}}$ represent the apsidal motion contributions of the gravitational, general relativistic and tidal effects, respectively.

Notes. (a) The units for the parameters are given in brackets. (b) For all the angular orbital elements: superscript $^{\mathrm{dyn}}$ refers to the dynamical coordinate system, while in the absence of any superscript, the symbol refers to the same orbital element in the observational frame of reference (for the difference of the two frames of reference, and the conversion between the two, see, e.g., Borkovits et al.[20]). (c) The superscript of "inf/sup" indicates inferior vs. superior conjunctions. (For the innermost orbit, we give the inferior conjunction, while for the medium orbit, a superior conjunction time is given).

Knowing the physical radii of components Aa and Ab, and considering that the innermost orbit is practically circular ($e_{\mathrm{in}} = 0.0074 \pm 0.0002$), one can calculate the projected equatorial velocities of both stars, assuming synchronous and coplanar rotation. We find $(v_{\mathrm{rot}} \sin i)_{\mathrm{Aa}}^{\mathrm{sync}} = 46$ km s$^{-1}$ and $(v_{\mathrm{rot}} \sin i)_{\mathrm{Ab}}^{\mathrm{sync}} = 23$ km s$^{-1}$ for the two components of the EB. The agreement between these results and the spectroscopically measured projected RVs (see in Results subsection Discovering and revealing the quadruple nature of TIC 120362137) makes it very likely that the components of the innermost EB rotate synchronously.

Turning to star B, which produces the third-body eclipses, its mass is found to be between the masses of the inner EB members ($M_{\mathrm{B}} \simeq 1.48\,M_\odot$). This star is still located on the main sequence and, therefore, more closely resembles the less massive star Ab than the already evolved, slightly more massive primary component, Aa. Star B has $R_{\mathrm{B}} \simeq 1.76\,R_\odot$ and $T_{\mathrm{B}} \simeq 6935$ K, and the latter value makes this component the hottest star in this quadruple. Regarding the fourth, outermost component, which we found to be the least massive star, this resembles our Sun with its mass of $M_{\mathrm{C}} \simeq 1.0\,M_\odot$, while its radius and effective temperature are found to be $R_{\mathrm{C}} \simeq 0.93\,R_\odot$ and $T_{\mathrm{C}} \simeq 5770$ K, respectively. The much higher uncertainties of these latter two quantities come from the fact that the radius and effective temperature of the fourth star have no effects on the light curve and RV curve solutions, and also have only a very limited effect on the net Spectral Energy Distribution (SED). Therefore, these properties (i.e., radius and temperature of the fourth star), are primarily computed from the evolutionary models built into the code, without any real observational constraints.

Regarding the geometric and dynamical parameters of the inner, triply eclipsing triple star subsystem, as well as the entire quadruple system, one may find it to be very "tight", and almost flat, i.e., coplanar. Triple systems are said to be "tight", when the gravitational third-body perturbations are well observable, which occurs for $P_{\mathrm{out}}/P_{\mathrm{in}} \lesssim 50$. For TIC 120362137, these ratios are $P_{\mathrm{mid}}/P_{\mathrm{in}} \lesssim 15.6$ (for the inner triple) and $P_{\mathrm{out}}/P_{\mathrm{mid}} \lesssim 20.4$. As to the flatness, the mutual inclination of the innermost and medium orbital planes is $(i_{\mathrm{mut}})_{A-B} = 0.47 \pm 0.26°$, that is, the inner triple subsystem remains flat, certainly to within 1°. Regarding the plane of the outermost orbit, in the absence of fourth-body eclipses, and due to the narrow observational window of a few years, we were only able to determine the corresponding mutual inclinations with much higher uncertainties, as $(i_{\mathrm{mut}})_{A, B-C} = 6 \pm 5°$. Even in this case, however, we may state that all three orbital planes remain well aligned, below 10°, with great likelihood. Such a flatness is very likely primordial, that is, a residual of the formation process of the entire quadruple system. The most likely scenario is that all four stars were formed from the very same, originally flat, disk with a sequential fragmentation process (see the review of Tokovinin[1]). Such a process, which favors the formation of nearly equal mass binary or multiple star components, and which had to be followed with some disk-driven migration, resulting in more compact inner subsystem(s) (as is the case in our current system), is reviewed in Offner et al.[29] (see also the modeling of Tokovinin and Moe[9]).

In Table 2 the fractional flux contribution of each star, as well as the contaminating extra flux, is given for all passbands. Here, we call attention to the very high fraction of the contaminating extra flux in the TESS passband. This contamination is due to the nearby (27″ away) bright star, TIC 120362128 (1 magnitude brighter), and the large (21″) TESS pixels. The same contamination does not affect the ground-based observations, as this nearby bright star was kept out of the photometric apertures of the ground-based telescopes (see also "Methods" subsection Space-borne and ground-based observations). Omitting, however, the contaminating light, one finds that the flux ratios of the four stellar components in the TESS-passband are 57.6, 15.4, 23.4, and 3.6% for components Aa, Ab, B, and C, respectively. These values are in nice accord with light ratios obtained from TRES

**Table 2 | Astrophysical parameters of TIC 120362137 from the joint photodynamical lightcurve, ETV, RV, SED, and PARSEC isochrone solution**

| stellar parameters | | | | |
|---|---|---|---|---|
| | **Aa** | **Ab** | **B** | **C** |
| **Relative quantities** | | | | |
| fractional radius [$R/a$] | $0.2207^{+0.0014}_{-0.0014}$ | $0.1110^{+0.0010}_{-0.0010}$ | $0.0183^{+0.0005}_{-0.0004}$ | $0.00120^{+0.00008}_{-0.00006}$ |
| temperature relative to $(T_{\rm eff})_{\rm Aa}$ | 1 | $1.0175^{+0.0035}_{-0.0034}$ | $1.0494^{+0.0042}_{-0.0046}$ | $0.8734^{+0.0243}_{-0.0232}$ |
| fractional flux [in TESS-band] | $0.2815^{+0.0047}_{-0.0048}$ | $0.0755^{+0.0008}_{-0.0008}$ | $0.1146^{+0.0061}_{-0.0056}$ | $0.0175^{+0.0043}_{-0.0032}$ |
| fractional flux [in Cousins $R_C$-band] | $0.5490^{+0.0131}_{-0.0136}$ | $0.1495^{+0.0033}_{-0.0034}$ | $0.2304^{+0.0135}_{-0.0114}$ | $0.0326^{+0.0083}_{-0.0063}$ |
| fractional flux [in Sloan $z'$-band] | $0.5697^{+0.0101}_{-0.0107}$ | $0.1520^{+0.0026}_{-0.0028}$ | $0.2280^{+0.0104}_{-0.0098}$ | $0.0380^{+0.0085}_{-0.0065}$ |
| Physical Quantities | | | | |
| $M$ [$M_\odot$] | $1.748^{+0.015}_{-0.018}$ | $1.361^{+0.012}_{-0.014}$ | $1.483^{+0.022}_{-0.022}$ | $1.008^{+0.047}_{-0.042}$ |
| $R$ [$R_\odot$] | $2.996^{+0.024}_{-0.025}$ | $1.507^{+0.017}_{-0.017}$ | $1.768^{+0.048}_{-0.041}$ | $0.927^{+0.062}_{-0.049}$ |
| $T_{\rm eff}$ [K] | $6612^{+52}_{-58}$ | $6724^{+50}_{-48}$ | $6937^{+61}_{-65}$ | $5772^{+171}_{-163}$ |
| $L_{\rm bol}$ [$L_\odot$] | $15.42^{+0.55}_{-0.64}$ | $4.17^{+0.17}_{-0.17}$ | $6.51^{+0.43}_{-0.41}$ | $0.86^{+0.24}_{-0.17}$ |
| $M_{\rm bol}$ | $1.80^{+0.05}_{-0.04}$ | $3.22^{+0.05}_{-0.04}$ | $2.74^{+0.07}_{-0.07}$ | $4.94^{+0.24}_{-0.27}$ |
| $M_V$ | $1.76^{+0.05}_{-0.04}$ | $3.19^{+0.05}_{-0.04}$ | $2.70^{+0.07}_{-0.07}$ | $4.99^{+0.27}_{-0.29}$ |
| $\log g$ [dex] | $3.726^{+0.005}_{-0.005}$ | $4.215^{+0.006}_{-0.006}$ | $4.113^{+0.015}_{-0.017}$ | $4.506^{+0.029}_{-0.037}$ |
| Global system parameters | | | | |
| $\log({\rm age})$ [dex] | $9.189^{+0.013}_{-0.011}$ | | | |
| [$M/H$] [dex] | $0.036^{+0.037}_{-0.038}$ | | | |
| $E(B-V)$ [mag] | $0.036^{+0.012}_{-0.012}$ | | | |
| extra light $\ell_x$ [in TESS-band] | $0.5102^{+0.0068}_{-0.0071}$ | | | |
| extra light $\ell_x$ [in Cousins $R_C$-band] | $0.037^{+0.022}_{-0.025}$ | | | |
| extra light $\ell_x$ [in Sloan $z'$-band] | $0.0087^{+0.0120}_{-0.0064}$ | | | |
| $(M_V)_{\rm tot}$ | $1.16^{+0.05}_{-0.05}$ | | | |
| distance [pc] | $584^{+7}_{-6}$ | | | |

The meaning of each parameter[a] is as follows: $M$ is the stellar mass; $R$ the stellar radius; $T_{\rm eff}$ the effective temperature; $L_{\rm bol}$ the bolometric luminosity; $M_{\rm bol,V}$ the absolute bolometric and visual magnitudes; $\log g$ the logarithm of the surface gravity (in cgs units); [$M/H$] the logarithmic metallicity abundance to H, by mass; $E(B-V)$ the color excess in B-V bands, $\ell_x$ the contaminating flux in the given band; and, $(M_V)_{\rm tot}$ is the absolute visual magnitude of the entire system.

Note. (a) The units for the parameters are given in brackets.

spectroscopy, using the QUADCOR algorithm (see above, in Results subsection Discovering and revealing the quadruple nature of TIC 120362137).

## Comparison with other 3 + 1-type compact quadruples

According to the latest version of the Multiple Star Catalog (MSC)[30], as well as to an extensive literature search, only two such compact 3 + 1-type quadruple star systems were previously known, of which the parameters are more or less comparable with the current system. These are HIP 41431[31] and TIC 114936199[32]. The former system, composed of four nearby, less massive stars, was discovered independently via its triple-lined spectroscopic nature and the cyclic ETVs of its innermost EB during three campaigns of the second or extended *Kepler*-mission (K2) observations[33]. Apart from the lower stellar masses, ranging from 0.35 to 0.63 $M_\odot$, this quadruple looks to be the closest analog of the currently investigated object TIC 120362137, at least from a dynamical point of view. This is so, on the one hand, because of the similar periods (and hence, period ratios), which are $P_{\rm in} = 3.28$ d vs 2.93 d, $P_{\rm mid} = 51.3$ d vs 59.2 d, and $P_{\rm out} = 1046$ d vs 1441 d for TIC 120362137 and HIP 41431, respectively. Moreover, the inner triple subsystems of both objects are formed by nearly similarly massive stars, while the fourth components are substantially less massive. In this regard, however, we note some important, but mainly only technical differences. While in the current quadruple system, the weak spectral lines of the less massive fourth star were directly detected, in the case of HIP 41431, the less massive outer component remains

undetected. Its presence was inferred only indirectly (though robustly), through the cyclic variations of the systemic RV of the inner triple subsystem (using several decades-long archival RV observations), and through the longer-term ETVs of the innermost EB. Moreover, another difference is that, though the inner triple subsystem was also found to be almost flat with $i_{\rm mut} = 2.2°$, the third star in HIP 41431 does not exhibit third-body eclipses.

In contrast to this, the outermost, fourth star in TIC 114936199 was identified through a 12-day-long extra eclipse sequence produced by the mutual eclipses between that fourth star and the innermost EB. We note that this is the only 3 + 1 type quadruple system, where the fourth star takes part in eclipsing events. Furthermore, from a morphological and dynamical point of view, this quadruple is less similar to TIC 120362137 than is HIP 41431. This is so because, in this quadruple, while the two inner periods are similar ($P_{\rm in} = 3.31$ d and $P_{\rm mid} = 51.2$ d), the outermost or fourth-body period appears to be substantially longer, and it is about $P_{\rm out} = 2100$ d. And, moreover, here the fourth component was found to be the most massive, while the two components of the innermost EB were found to be two very low mass red dwarf stars, and, therefore, they were not detected spectroscopically.

It is very likely that there exist a number of other, similarly compact, 3 + 1 type quadruple stellar systems. Their discovery is, however, very difficult and may depend on only some occasional, fortuitous properties of these systems. It is easiest if the innermost binary is an EB. Then, its ETV may quickly indicate a close, tight third stellar component. Verifying the existence of a more distant, longer-period fourth

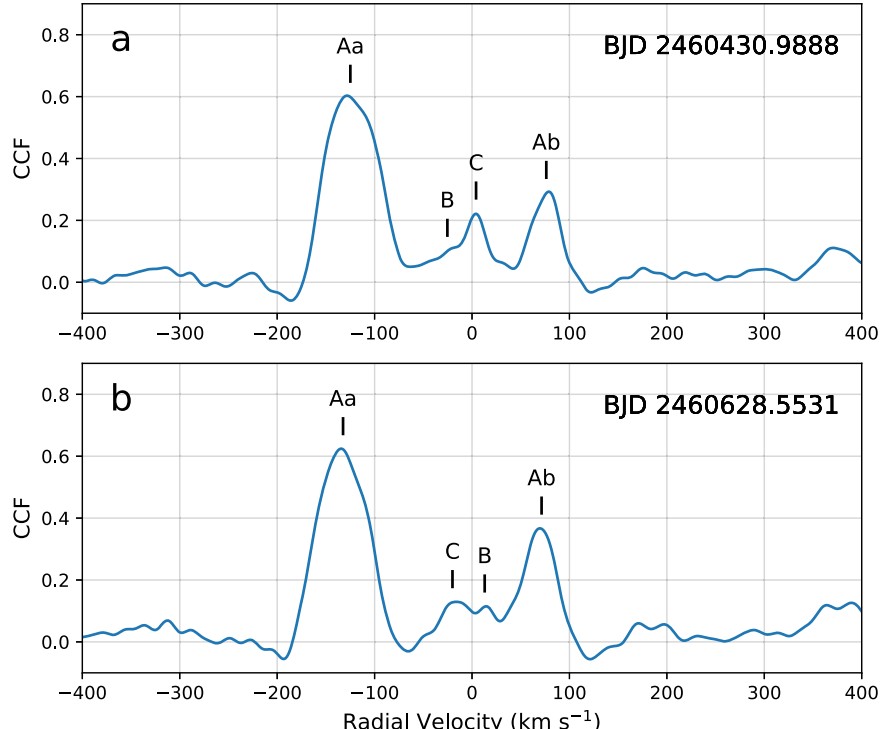

**Fig. 3 | One-dimensional cross-correlation functions (CCFs) for two spectra of TIC 120362137. a** TRES spectrum taken on BJD 2460430.966. **b** TRES spectrum taken on BJD 2460628.5531. Both spectra show all four component stars of the system. (Components Aa and Ab are the primary and secondary stars of the innermost, close binary. Component B is the more distant, third component star of the inner triple subsystem, while component C is the most distant, fourth star.) Note that the apparently low significance of the bumps representing star B in this 1-D CCF is somewhat misleading and is caused by its very rapid rotation, which lowers the contrast. Component B is actually the 2nd brightest star in the system (twice as bright as Ab, and almost 15 times brighter than C). The figure does not necessarily reflect the reliability of the results from QUADCOR, which operates in higher dimensions. This allows QUADCOR to significantly reduce the deleterious effects of blending with the lines of the other stars, isolating each component and making it easier to measure RVs. Source data are provided as a Source data file.

star, however, requires rare and accidental circumstances, such as the presence of fourth-body eclipses or longer timescale spectroscopic or photometric follow-up observations. In this regard, it should be noted that Zasche et al.[34], Hajdu et al.[35], and others have been reporting some doubly periodic ETVs in the case of some EBs, which might indicate additional compact quadruple systems of the 3+1 kind, but these findings are very uncertain and will need further continuous monitoring.

In conclusion, according to our knowledge, there are no other known, similarly compact and tight, planetary-system-like 3 + 1 quadruple stellar systems. At the time of writing this paper, TIC 120362137 is not only the most compact amongst them, but also the only one in which all four stellar components were detected directly through spectroscopic observations.

## Discussion

The dynamical stability of tight triple and multiple stellar systems is a long-standing problem. No strict analytical formulae are known which can uniquely predict the stability of any given system as a function of its dynamical parameters (masses and orbital elements). One of the currently most accepted and widely used semi-empirical formula for the likely long-term dynamical stability of a tight triple stellar system is that of Mardling & Aarseth[36], which was slightly modified and improved, e.g., by Vynatheya et al.[37]. This stability criterion was originally found for the ratio of the pericenter distance of the outer orbit and the semi-major axis of the inner orbit, however, it can be easily reformulated in terms of the period ratios of the outer to the inner orbits (which are the most easily measurable parameters) and, this

results in:

$$
\left(\frac{P_{\text{out}}}{P_{\text{in}}}\right)_{\text{stab}} \gtrsim 4.69(1+q_{\text{out}})^{1/10}\frac{(1+e_{\text{out}})^{3/5}}{(1-e_{\text{out}})^{9/5}}\left(1-\frac{0.3i_{\text{mut}}}{\pi}\right)^{3/2}, \quad (1)
$$

where $P_{\text{in,out}}$ denote the orbital periods of the inner and outer subsystems, respectively; $q_{\text{out}}$ and $e_{\text{out}}$ stand for the mass ratio and the eccentricity of the outer subsystem, while $i_{\text{mut}}$ represents the mutual inclination angle of the two orbital planes.

Substituting the corresponding parameters from Table 1 into this expression, one finds this indicates that the inner 2 + 1-type triple subsystem is expected to be stable for orbital periods $P_{\text{mid}} \gtrsim 28.48$ d, while the outermost subsystem (which is also considered in this regard to be a triple system), would be stable for $P_{\text{out}} \gtrsim 498.1$ d. Both of these conditions are fulfilled very robustly in our quadruple system, and therefore, one can expect that TIC 120362137 should be dynamically stable, at least for its full main-sequence (MS) lifetime.

A weakness of this criterion, however, is that, as was mentioned, it is only a semi-empirical and statistical formulation that was obtained with the use of many numerical integrations of multiple point mass models. Moreover, as was shown in Vynatheya et al.[38], real 3 + 1 type quadruples in many cases show different stability properties than would be expected when they are approximated simply by two triples, as was done above. One should keep in mind, however, that the system is empirically found to be dynamical stable for more than one billion years (1 Gyr) based on its current age.

The future evolution of this system (see Supplementary Figs. S1 and S2) has also been explored using the stellar evolution code

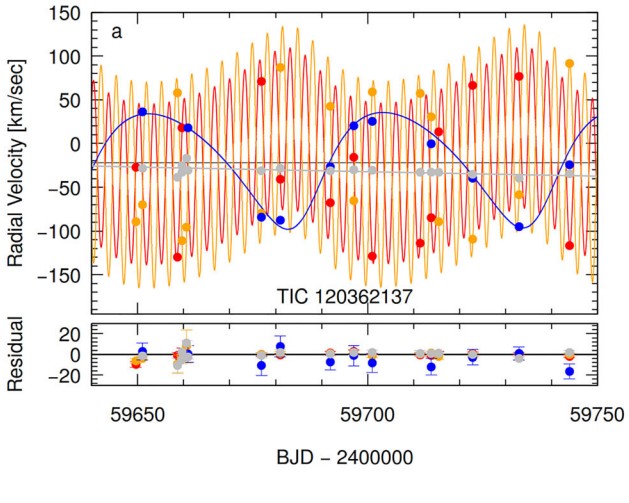

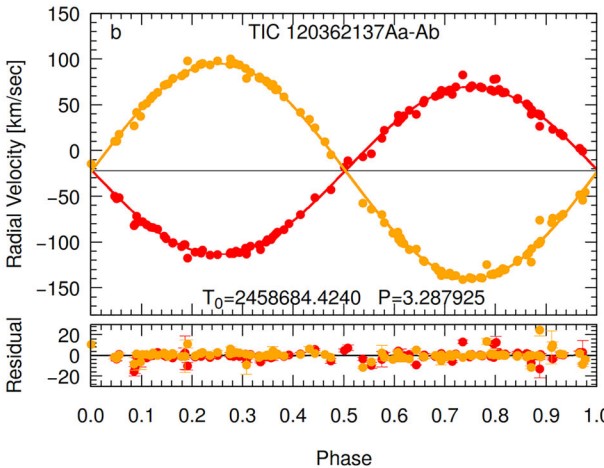

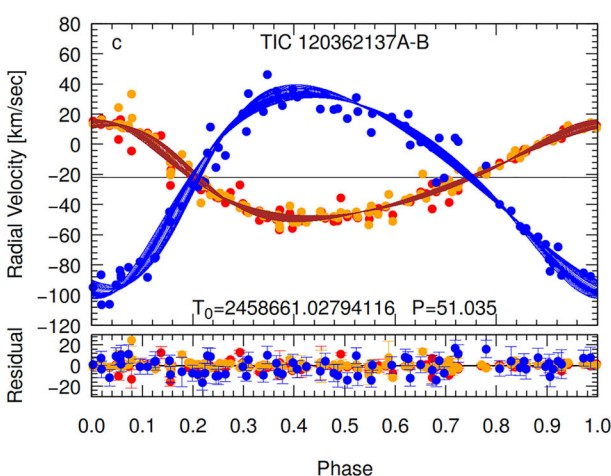

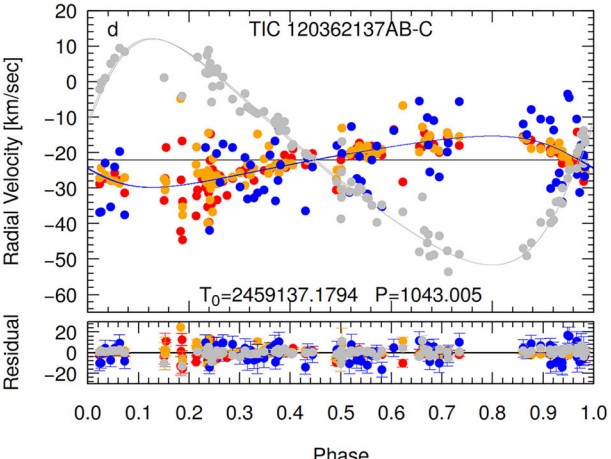

**Fig. 4 | RV curves and model fits for TIC 120362137. a** A characteristic, nearly 4-month-long section of the RV curves in the time domain. **b** RV curves folded by the shortest period orbit, i.e., that of the innermost binary. **c** RV curves folded by the period of the medium orbit. **d** RV curves folded by the period of the outermost orbit. Red, orange, blue, and gray dots represent the observed RV values of components Aa, Ab, B, and C, respectively. In the case of each of the folded RV curves, we removed the RV contributions of the other two orbits, while the more or less continuous lines represent the RV values calculated from the best spectro-photodynamical solution. The thin, horizontal line at $V = -22.12$ km s$^{-1}$ represents

the systemic radial velocity of the quadruple star system. Note that the very apparent thickness of the model-solution line(s) in the case of the approximately 51-day-period, folded middle orbit (120362137A-B) panel (that is, **c**), does not come from an incorrect folding, but rather is the consequence of the non-Keplerian motion. Primarily, it is due to the rapid, dynamically forced apsidal motion of the eccentric, middle subsystem. Observed minus modeled residual curves are plotted in the lower parts of each panel. In these residual curves, the error bars represent the uncertainties of each individual measurement. Source data are provided as a Source data file.

Modules for Experiments in Stellar Astrophysics (MESA, version 12115)[39–43]. While that statement is certain, that the primary star of the inner EB (that is, star Aa) will the first one which will fill out its Roche-lobe (that is, the region surrounding such a star, which is a member of a binary system, within which the material can remain bound to the star) any further investigations need certain reasonable assumptions, and therefore, any such projections into the future carry with them a degree of uncertainty. Therefore, we discuss such a likely, but not fully certain, evolutionary study in the Supplementary Information part.

## Methods
### Space-borne and ground-based observations
TIC 120362137 is a relatively bright (about 10th magnitude) star in the northern, summer sky, in the constellation Cygnus, just 3–4° below the original field of the *Kepler* space telescope. TESS observed this target during nine sectors between 2019 and 2024. Each observing sector was 27–28 days long. Our target was observed only in full frame image (FFI)

mode during all these occasions. The cadence times were 1800 s (for Sectors 14 and 26), 600 s (Ss 40, 41, 53, 54), and, finally, 200 s (Ss 74, 80, 81). It should be noted, however, that due to the large pixel size of the TESS cameras, the present target was strongly contaminated by the nearby star, TIC 120362128, which was observed in 2-min cadence mode during the same sectors. Therefore, indirectly, we were able to use 2-min cadence light curves for the determination of the mid-eclipse times, since the signal of our target was clearly visible in this latter set of observations. Regarding FFI data, with the exceptions of the last two sectors of observations, we processed the lightcurves using a convolution-based differential photometric pipeline implemented in the FITSH package[44]. The 2024 summer sectors (80–81) were processed, however, with the use of the publicly available pipeline LIGHTKURVE[45].

Photometric follow-up observations were carried out with similar 80-cm Ritchey-Chrétien (RC) telescopes of the Baja Astronomical Observatory (BAO80) and the Gothard Astrophysical Observatory

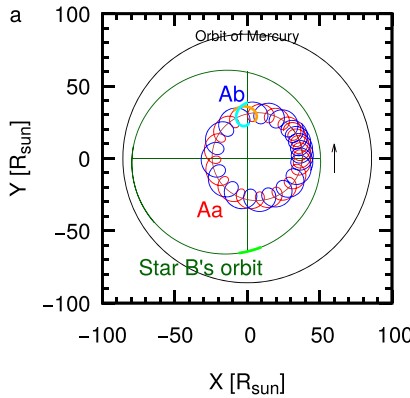
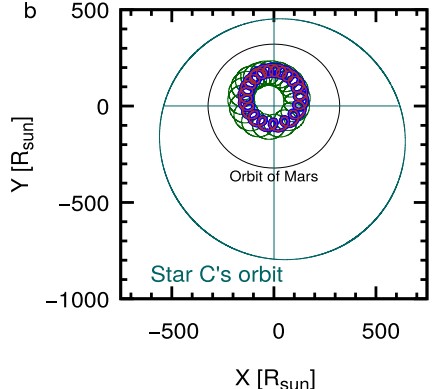

**Fig. 5 | Illustration of the architecture and true physical dimensions of the compact quadruple star system, TIC 120362137. a** The orbits of the inner triple subsystem (formed by the innermost EB, stars Aa and Ab, and star B) during one revolution of star B around the center of mass of this triple (fixed at point 0,0). The orbits are drawn in red (star Aa), blue (Ab), and green (B), respectively. Moreover, the orbit of the center of mass of the innermost binary is also plotted in brown, while the short upward arrow represents the orbital direction. Note, the observer (Earth, or TESS) is located along the positive $Y$ axis. The highlighted small arcs of the orbits of the three stars denote those sections where the components were located during the first TESS-observed third-body eclipse in Sector 14. For comparison, the

average distance of Mercury from our Sun is also noted with a thin black circle. **b** The orbits of the entire quadruple during one revolution of the outermost component, C (dark green). The orbits of the other components plotted with the same colors as on the left panel. (Note, however, that the red orbits of component Aa are largely hidden (overdrawn) by the blue colors of star Ab.) The average Sun–Mars distance is also shown with a thin black circle. The extra cyclic behavior seen in the A and B orbits is due to the superpositions of three (the innermost, middle, and outermost) and two (the middle and the outermost) Keplerian orbits, respectively. Source data are provided as a Source data file.

(GAO80) in Hungary. Details of these instruments and observations are described in Borkovits et al.[46]. The BAO80 data were collected on ten nights between 2021 June and 2024 May, with the use of a Sloan $r'$ filter, while the GAO80 data cover 22 nights between 2021 March and 2024 September in the Sloan $z'$-band. Moreover, on five nights the star was observed at the Ondřejov Observatory, in the Czech Republic, using a 65-cm telescope and standard Cousins $R_C$-filter[47]. An additional 19 nights of observations were carried out with smaller aperture (20 cm/15 cm, and 34 mm) refractors, equipped with an R filter or using unfiltered photometry. All of the latter are located in small private observatories, which are in the Czech Republic, and named "Veltěže u Loun", "Jílové u Prahy", and the BS Observatory at Zlin. Moreover, the target was also observed on one night with an 80-cm telescope equipped with a C5A-150M camera (Sloan $r'$ filter) of the Žďánice Observatory, in the Czech Republic. These Czech observations were obtained between 2021 April and 2024 September. Finally, the target was also observed with a 50-cm RC telescope at Patterson Observatory, Arizona. For the description of this instrument, see also Borkovits et al.[46]. The datasets obtained from all these observations are provided in Supplementary Data 2–5.

Spectroscopic measurements were obtained with four different instruments. A total of 73 spectra were collected between 2020 September and 2024 November at the Center for Astrophysics, using the TRES Spectrograph on the 1.5 m Tillinghast reflector located at the Fred L. Whipple Observatory on Mount Hopkins (Arizona, USA). The instrument delivers a resolving power of approximately $R = 44{,}000$, and records 51 echelle orders over the wavelength range 3800–9100 Å. The spectra that we obtained had signal-to-noise ratios ranging from about 35 to 100 per resolution element of 6.8 km s$^{-1}$. Reductions were carried out with a dedicated pipeline (see Buchhave et al.[48]), and wavelength solutions relied on exposures of a thorium-argon lamp taken before and after each science exposure. Instrumental shifts were monitored by observing standard stars during each run, and asteroid observations were used to transfer the velocities onto the International Astronomical Union (IAU) system.

An additional 20 spectra were obtained with the MUlti-SIte COntinuous Spectroscopy (MUSICOS)-clone échelle spectrograph fiber-fed from the 1.3 m, f8.36 Nasmyth-Cassegrain telescope at the Skalnaté Pleso Observatory (Slovakia). The instrument has maximum resolving

power around $R = 38{,}000$ and records 56 échelle orders covering the 4240–7250 Å wavelength range. The spectra are typically a combination of three 900-s exposures to boost the Signal-to-Noise Ratio, which ranged between 16 and 63 per pixel (about 3 km s$^{-1}$). The reductions were done by a dedicated pipeline under IRAF using shell scripts. The wavelength system was derived from the ThAr hollow-cathode lamp spectra taken before and after object exposures. The typical RV stability of the spectrograph was about 200 m s$^{-1}$.

Moreover, the target was also observed by the 1-m Ritchey-Chrétien-Coudé (RCC) telescope of Piszkéstető Mountain-station of Konkoly Observatory, Hungary, and these measurements resulted in 26 additional spectra on 25 nights between 20 March 2021 and 30 August 2024. The telescope has an échelle spectrograph with about $R = 20{,}000$ that covers the wavelength range of 3890–8670 Å in 33 échelle orders. It uses a back-illuminated FLI ML1109 CCD camera with an array of $2048 \times 506$ pixels of size 12 μm, 10 e$^-$ readout noise, and a gain close to unity. Exposure times varied between 3600 and 7200 s according to the actual sky quality. Finally, 2 spectra were also obtained with the 2-m Rozhen telescope equipped with an approximately $R = 30{,}000$ échelle spectrograph on the nights of 11 May 2022 and 8 September 2022. Additional details of this instrument can be found in Bonev et al.[49]. All of these spectra were reduced with a pipeline utilizing a combination of IRAF[50,51] and iSpec[52,53] methods, including the steps of bias, dark, and flat field corrections, wavelength calibration, continuum normalization, telluric line removal, and barycentric correction.

Both the photometric and spectroscopic follow-up observations are summarized in Supplementary Tables S1 and S2.

RVs were extracted from the non-TRES spectra with the code BF-rvplotter (https://github.com/mrawls/BF-rvplotter), which we used to calculate the broadening functions (BFs) in a ±300 km s$^{-1}$ velocity range using 3 km s$^{-1}$ velocity bins and a Gaussian smoothing with a rolling window of five data points. The BFs were calculated using the 4800–6500 Å wavelength range, and iSpec's built-in spectrum of the Sun was used as a template. The resulting BFs were modeled by a sum of three Gaussians to find the individual RVs of the components.

RVs for the four components from the TRES data, however, were derived by cross-correlation, as described in more detail above, in the Results subsection Discovering and revealing the quadruple nature of

TIC 120362137. Templates for each star were taken from a large library of pre-computed synthetic spectra, which are based on model atmospheres by R. L. Kurucz[54] and a line list tuned by hand to improve the match to real stars. These templates cover a wide range in effective temperature ($T_{eff}$), surface gravity ($\log g$), rotational broadening ($v \sin i$), and metallicity. The templates span a narrow wavelength window centered on the region of the Mg I b triplet near 5187 Å, which experience has shown captures most of the velocity information. For the present work, we assumed solar metallicity and surface gravities from a preliminary global analysis, rounded to the nearest $\log g$ value in our template library. These values did not change with the final global analysis described later.

### Spectro-photodynamical analysis

We extracted the stellar and orbital parameters of our 3 + 1 quadruple system by using a sophisticated spectro-dynamical analysis. For this key part of the analysis, we utilized the software package LIGHT-CURVEFACTORY (see, e.g., Borkovits et al.[55], and references therein). This code does or contains the following things. (1) Given a trial set of orbits and stellar parameters, it emulates (multi-passband) lightcurves, ETV curves, and RV curves (if the latter exist). (2) There are built-in tables of Padova and TRieste Stellar Evolutionary Code (PARSEC) evolution tracks and isochrones (Bressan et al.[56]), which can be used to represent stellar evolution models (in particular to estimate radii and $T_{eff}$ values given the stellar mass, age, and metallicity). In turn, this enables the net SED of multi stellar systems to be computed. (3) The code has a numerical integrator that calculates the complex orbital motions of two to six bodies. This utilizes a seventh-order Runge–Kutta–Nyström algorithm. (4) The best-fitting system parameters are found with an MCMC-based search routine that also computes the statistical uncertainties. The jump condition makes uses our own implementation of the Metropolis–Hastings algorithm[57].

LIGHTCURVEFACTORY has been developed over the past decade to extract system parameters from multiple star systems up to six constituent stars. For four-body systems, the two different hierarchies of 2 + 2 and 3 + 1 stars are handled separately. The details of how the code functions, the analysis procedure, and the code's use in investigating a broad range of binaries, triples, quadruples, and even a sextuple system are described in numerous papers[31,46,55,58].

In the current analysis, we simultaneously fit the following photometric and spectroscopic observational data:

1) Light curves in three different bandpasses. The longest duration and most accurate dataset comes from the photometric observations of TESS. Similar to the other two sets of photometric observations (see below), the light curves derived from TESS observations were converted into normalized flux values, where the unit flux value represents the mean flux during the out-of-eclipse sections. In order to reduce computational time, we converted all the available TESS light curves into 1800 s bins, which still yield 15–17 data points across the eclipses. Again, to further reduce the computational time, we dropped out the mostly inconsequential out-of-eclipse light curve sections, and kept only photometric phase sections of $0.9 \leqq \phi_{Aab} \leqq 1.1$ and $0.41 \leqq \phi_{Aab} \leqq 0.61$. Moreover, naturally, we left the light curve sections with third-body eclipses in our analysis as well. This final TESS dataset, which was used for our analysis, is provided as Supplementary Data 1.

   Turning to the ground-based photometric follow-up observations, which originally aimed only to get further times of eclipsing minima for the ETV curves, we decided to include them also in the entire light curve analysis. The GAO80 observations (that are provided as Supplementary Data 2) were carried out in the Sloan $z'$-band; therefore, we modeled these data by setting this photometric band in the code. The other observations were carried out in Sloan $r'$ or Cousins $R_C$ bands. Despite the inhomogeneity of these observations, we modeled the latter two together as Cousins $R_C$-band observations (that are provided as Supplementary Data 3). As these ground-based observations were gathered with different exposure times, even with the same instruments, due to the varying sky qualities night by night, we formed 900 s bins from each data set. (These two datasets are provided as Supplementary Data 4 and 5.) These latter light curves got ten times lower weight per point than that of the TESS light curve, and their only significance during the photodynamical analysis comes from the fact that, in contrast to the TESS data, these are not contaminated with the flux of the nearby, bright star TIC 120362128 (as this latter object was kept out of the apertures of these ground-based observations). Therefore, these light curves help to give more realistic estimations for the true value of any additional extra flux in the system.

2) ETV curves derived from the photometric observations. We derived mid-eclipse times for all observed primary and secondary eclipses of the innermost EB. In order to do this, we folded and averaged all the available eclipse light curves to form template primary and secondary eclipse light curves. Then these templates were fitted to each individual eclipse section. A more detailed description of the method can be found in Borkovits et al.[59]. Such ETV curves provide the primary information for most of the dynamical parameters of a compact triple or, in the current case, quadruple star system. Naturally, as the times of minima are derived from the light curves, which are now included in our analysis, fitting ETV curves separately (but simultaneously) may appear at first sight to be redundant. These curves, however, contain an extract of all the dynamical information, and they are much more sensitive to even small departures in the periods and some other orbital elements than the entire eclipsing light curves. Therefore, they lead to a much faster and more robust convergence of the whole process. These ETV data are tabulated in Supplementary Table S3, and are also provided in machine-readable form as Supplementary Data 6.

3) RV curves of all four component stars. As was mentioned above, the spectral lines of all four stellar components were detected separately, and, therefore, it was possible to determine individual RVs for each star. Or, in other words, this system is a four-lined spectroscopic quadruple system. Consequently, we included four separate data files into our analysis, containing data of time-RV-error triplets for each star. One should note that, despite the fact that we obtained several RV data sets with four different instruments, the presented analysis is mainly based on the TRES data (see Supplementary Table S4). This is so because the faint lines of the fourth component were primarily detected only with this instrument. Therefore, since the TRES RV data covers in time the motion of all four components well, and they are sufficient for the entire analysis by themselves, we decided to avoid some inhomogeneities that may occur when one combines different sets of RV data. Thus, to obtain the first quantitative results, we used only TRES data. Then, however, we added RV data obtained with the other instruments, and found that there are no detectable offsets or systemic RV-velocity shifts in these data. Hence, finally, we repeated the analysis using all the RV data. In this regard, however, one should note that, after obtaining the first quadruple-star system solutions, we found that the weakest spectral lines in the non-TRES spectra, which we first suspected to be signals from the third component, B, might instead belong to either component C or be only artefacts. Therefore, in the final analysis mentioned above, we handled the RVs determined from these lines accordingly, and they are also listed as RV-s of

component C in Supplementary Tables S5 and S6. Note, all the mentioned and used RV data are also provided in machine-readable format in Supplementary Data 7–9.

4) Catalog passband magnitudes for composite SED analysis. The measured magnitude values from UV to IR were also gathered, and fitted them against theoretically computed values. While all of the above three items are independent of any astrophysical models (apart from the templates used to fit spectra for RV data), calculating theoretical magnitudes for the SED fitting, the current code uses theoretical PARSEC isochrones, which makes the presented model also dependent on the astrophysical model which was applied to calculate theoretical PARSEC isochrones and stellar magnitudes. These isochrones are tabulated in the code in grids of (metallicity, age, and stellar mass) triplets. The current values of the astrophysical parameters to be constrained (see below) are selected and calculated through 3-dimensional interpolation from neighboring grid-points. A more detailed description of the entire process can be found in Borkovits et al.[55].

In conclusion, during MCMC trial runs, we fit the following parameters:

i) The initial values of three of six orbital element-related parameters of the innermost, and six of six parameters of the middle and outermost orbits. The three parameters that are fitted in the case of all three orbits are as follows: the observable inclinations ($i_{in,mid,out}$), and the two components of the eccentricity vectors ($e \sin \omega$)$_{in, mid, out}$, ($e \cos \omega$)$_{in, mid, out}$. The other parameters, which apply only to the middle and the outer orbits include the periods ($P_{mid,out}$), and the nodes of these two orbits ($\Omega_{mid,out}$) relative to the node of the innermost orbit, the latter of which is set to $\Omega_{in} = 0$. Finally, in the case of the middle orbit, we fit for the time of the (middle of the) first observed third-body eclipse or, more strictly speaking, the time of the first superior conjunction of the third star on its orbit around the common center of mass of the inner triple subsystem ($\mathcal{T}_{mid}^{sup}$) and, in the case of the outer orbit, in the absence of such an observable as the mid-time of a (fourth-body) eclipse, the time of a periastron passage ($\tau_{out}$). One should keep in mind, however, that these are instantaneous, osculating orbital elements, which are valid only at and for the given epoch $t_0$, and they are subject to variations with time due to the gravitational perturbations.

ii) Four mass-related parameters: the mass of the most massive component, that is, the primary, Aa, of the innermost EB, and the three mass ratios, ($q_{in,mid,out}$);

iii) The only freely adjusted light curve-related parameter is the passband-dependent extra flux ($\ell_x$), which should be set independently in all three passbands, and finally

iv) Those parameters that are connected to the SED fitting and the PARSEC isochrones, such as the logarithmic age (log $\tau$) and metallicities ([M/H]) of the entire quadruple system (these are considered as global parameters), and the extinction of the system $E(B − V)$.

Besides these adjusted parameters, some other variables were constrained, that is, computed and set automatically in and by the code, as follows:

i) The (initial, osculating) period of the innermost binary ($P_{in}$) and the time of an inferior conjunction of the secondary, Ab, relative to Aa (that is, the mid-time of a primary eclipse) are calculated automatically at each step, iteratively minimizing the value of $\chi_{ETV}^2$;

ii) The systemic RV, $\gamma$, is calculated in the same manner, but strictly a posteriori at the end of each trial run, minimizing $\chi_{RV}^2$;

iii) The stellar radii ($R_{Aa-C}$), the effective temperatures ($T_{Aa−C}^{eff}$), as well as the individual passband magnitudes of each star are calculated from the built-in three-dimensional (mass, age, and metallicity) grids of PARSEC isochrones;

iv) Passband-dependent limb darkening coefficients—taking into account the currently applied limb-darkening law(s), calculated iteratively using the built-in grids of Prša and Zwitter[60];

v) Finally, the photometric distance of the target is also constrained a posteriori, minimizing $\chi_{SED}^2$ at the end of each trial step.

Finally, we comment on the reliability of the presented parameter determinations for the current quadruple system, obtained through the above-described process. The joint analysis of the photometric light curve, the ETV curve, the SED curve, and especially the RV data does not explicitly depend on any a priori assumption about the former evolutionary history of the current system, specifically any prior episode of mass transfer. Therefore, the orbits, stellar masses, radii, and temperature ratios are relatively immune to any possible previous mass transfer in the system. Even the absolute values of the temperatures are mainly constrained by the (cumulative) SED. On the other hand, however, when PARSEC isochrones are used, which (for this system) mainly constrain the metallicity and age of the stars, this naturally introduces some astrophysical model dependences into the analysis, and thus, some uncertainties in the age and metallicity, but not in the masses, radii, and $T_{eff}$ values. Therefore, we believe that the temperature uncertainties that we cite are realistic, and in accord with the introductory statements of Miller et al.[61].

## Data availability

This paper makes extensive use of data collected by the TESS mission. TESS observations can be downloaded freely from Mikulski Archive Space Telescope (MAST) Portal (https://mast.stsci.edu/portal/Mashup/Clients/Mast/Portal.html). The processed, binned light curves used for spectro-photodynamical analysis are provided here as Supplementary Data 1. Ground-based photometric light curves that we obtained, as well as their binned versions, are provided in the files Supplementary Data 2–5. Mid-eclipse times determined from these photometric observations are tabulated in Supplementary Table S3 and provided as machine readable plain ascii data in Supplementary Data 6. All the RV data obtained with the four spectrographs mentioned in the text are tabulated in Supplementary Tables S4–S6 and provided as machine readable plain ascii data in Supplementary Data 7–9. Moreover, each individual spectrum can be downloaded from Zenodo site using the following URL: https://zenodo.org/records/17914597. For the SED analysis, catalog passband magnitudes were obtained from the TIC-v8.2 and Gaia DR3 catalogs, which are available also in the MAST Portal (see above). Distance and other astrometric properties for our target were taken from the archives of European Space Agency (ESA) mission *Gaia* (https://www.cosmos.esa.int/gaia), processed by the *Gaia* Data Processing and Analysis Consortium (DPAC) (https://www.cosmos.esa.int/web/gaia/dpac/consortium). Source data are provided with this paper and include data Figs. 1a, b, 2, 3a, b, 4a–d, 5a, b, as well as Supplementary Figs. S3a–i and S4a–j. Source data are provided with this paper.

## Code availability

For the reduction of the freely available TESS photometric data FITSH[44] and LIGHTKURVE[45] softwares were used, which are available (together with detailed descriptions) in the following URLs: https://fitsh.net/and http://ascl.net/1812.013, respectively. Spectroscopic data were processed either with BF-RVPLOTTER (https://github.com/mrawls/BF-rvplotter) or QUADCOR[27]. This latter software is available upon reasonable request from the corresponding author. PARSEC isochrone tables, which are built into the software package LIGHTCURVEFACTORY, were generated with the use of data available at http://stev.oapd.inaf.it/PARSEC. Evolutionary studies were carried out with the MESA[39–43] package, which is downloadable and installable from the following site: https://docs.mesastar.org/en/latest/installation.html. Finally, the photodynamical analysis of all the available observational datasets was carried out with software package LIGHTCURVEFACTORY. The main properties of this software

is described, for example, in the papers:[31,46,55,58]. The current version is available upon reasonable request from the corresponding author, but after finalizing the graphical user interface that is currently being developed, a more user-friendly version of the software package will be publicly available.

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

## Acknowledgements

This project has received funding from the HUN-REN Hungarian Research Network. T.B., T.M., I.B.B., and A.P. acknowledge the financial support of the Hungarian National Research, Development and Innovation Office – NKFIH Grants K-147131 and K-138962. T.B. acknowledge also the support of the University of Szeged Open Access Fund, Grant Nr. 8437. H.-L.C. is supported by the National Key R&D Program of China (grant Nos. 2021YFA1600403), the National Natural Science Foundation of China (grant Nos. 12288102, 12333008, and 12422305), and the Chinese Academy of Sciences "Light of West China" Program. TP and ZG were supported by the VEGA grant of the Slovak Academy of Sciences No. 2/0033/26 and by the Slovak Research and Development Agency, the contract No. APVV-24-160. ZG and ZD acknowledge the support of the PRODEX projects between the ELTE Eötvös Loránd University and the European Space Agency, "Observation of Exoplanets with the CHEOPS Space Observatory" (PEA 4000137122), "Hungarian Contribution to ESA's Ariel Space Telescope Mission: II. High-Precision Photometry with Ariel" (PEA 4000149203), and "Hungarian Contribution to ESA's Ariel Space Telescope Mission: III. Stellar Characterisation" (PEA 4000149202), the support from SNN-147362 and the ADVANCED-153410 of the National Research, Development and Innovation Office (NKFIH, Hungary),and the support of the city of Szombathely. D.W.L. is acknowledged for his help in organizing the spectroscopic observations of TIC 120362137 at the Center for Astrophysics. Funding for the TESS mission is provided by the NASA Science Mission Directorate. Some of the data presented in this paper were obtained from the Mikulski Archive for Space Telescopes (MAST). STScI is operated by the Association of Universities for Research in Astronomy, Inc., under NASA contract NAS5-26555. Support for MAST for non-HST data is provided by the NASA Office of Space Science via grant NNX09AF08G and by other grants and contracts. Funding for the DPAC has been provided by national institutions, in particular the institutions participating in the *Gaia* Multilateral Agreement. Some of the SED fluxes and magnitudes were obtained with the Wide-field Infrared Survey Explorer, which is a joint project of the University of California, Los Angeles, and the Jet Propulsion Laboratory/California Institute of Technology, funded by the National Aeronautics and Space Administration. Additionally, some of the SED fluxes and magnitudes were obtained with the Two Micron All Sky Survey, which is a joint project of the University of Massachusetts and the Infrared Processing and Analysis Center/California Institute of Technology, funded by the National Aeronautics and Space Administration and the National Science Foundation. We used the Simbad service operated by the Centre des Données Stellaires (Strasbourg, France) and the ESO Science Archive Facility services (data obtained under request number 396301). This research has also made use of the VizieR catalogue access tool, CDS, Strasbourg, France (https://doi.org/10.26093/cds/vizier). The original description of the VizieR service was published in ref. 62.

## Author contributions

B.T. and S.R. organized the follow-up observations, determined the ETV points, made the photodynamical analysis, and wrote the paper. H-L.Ch. carried out the MESA analysis on the future evolution, G.T., T.M., and T.P. made spectroscopic observations, extracted the RV points, and made some preliminary RV analyses, B.P. and V.K. identified the triply eclipsing nature of this target and investigated the dynamical stability of the system, P.Z., I.B.B., Z.G., I.Cs., Z.D., D.C., J.K., P.Č, Z.H., T.K., H.K., M.M, and R.U. provided photometric follow-up data.

## Funding

## Competing interests

The authors declare no competing interests.
