## [Transparent Peer Review file · Nature Communications]

Discovery and analysis of the most compact 3+1-type quadruple stellar system, TIC 120362137

Corresponding Author: Dr Tamás Borkovits

Version 0:

Reviewer comments:

Reviewer #1

(Remarks to the Author)

To the authors:

Review of the Manuscript: Discovery of the most compact 3+1-type quadruple star system, TIC 120362137 by Tamas Borkovits et al, submitted to Nature Communications.

The manuscript reports the discovery and detailed observational analysis of TIC 120362137, claimed to be the most compact known 3+1 hierarchical quadruple star system. The authors combine photometric and spectroscopic data to characterize the system and discuss its evolutionary history and future.

Despite the manuscript's claim of being a "dynamical study," no direct N-body or secular N-body simulations are presented. The authors do not address the dynamical stability of the system, which is crucial for such a compact and hierarchical quadruple. Without such analysis, the long-term viability and evolutionary predictions for the system remain speculative and incomplete. In particular, such a study is also important for the $>Gyr$ stability of the system.

The introduction discusses the evolution of X-ray binaries, which is not relevant to the subject of this paper. This distracts from the focus and dilutes the impact of the manuscript.

The paper is unnecessarily long, with significant portions not directly relevant to the main discovery or analysis. This detracts from readability and clarity, and the manuscript would benefit from substantial condensation.

The discussion on the formation mechanism is not clearly applicable or relevant to the specific system under study. The formation scenario should be directly tied to the observed properties and uniqueness of TIC 120362137, rather than relying on generic or loosely connected arguments.

to this extend, the term "To the best of our knowledge" does not give sufficient warranty for original research. The authors should do their homework, and make sure that it either "is" or "is not" the most compact quadruple discovered so far. The point itself, however, is rather irrelevant. Maybe, it would be better if the authors just

ignore the entire mention of "first" and "best", and state the facts, rather than their desire for novelty.

The authors predict that the inner binary will undergo a common envelope (CE) phase upon the primary filling its Roche lobe, leading to a merger. However, this scenario is questionable. For such binaries, the first mass transfer episode is more likely to be stable and largely conservative, rather than leading to a CE phase. The mass transfer would cause the inner orbit to expand, as well as the orbits of the outer companions. Only a small fraction of the mass is typically lost from the system in these cases. This omission leads to an inaccurate depiction of the likely evolutionary pathway.

This discussion of the system's future, being highly uncertain, and depends critically on dynamical stability and the stability of mass transfer. In particular the latter is one of the least certain aspects of binary evolution. The manuscript would be stronger if it focused more on the past and formation of the system, which are more constrained by the data, rather than speculative future scenarios. How do one get four similarly-massive stars with very small mutual inclination in almost circular orbits. The use of a common stellar evolution code is no guarantee here. Common binary evolution theory will render this system stable against common envelope. The authors should at least provide a thorough analysis of the Darwin-Riemann (or other) instability criteria that initiates the CE.

Please, check for orbital resonance, in particular check to what extent the quadruple is in mean-motion resonance.

Terms such as "significant," "the most compact" (used 27 times, including in the title), and "for the first time" are overused. Such language should be reserved for clear, quantifiable distinctions and not repeated excessively, as it can undermine scientific objectivity.

The first author cites his own work 20 times out of ~60 references. This is excessive and inappropriate, especially given that the author does not dominate the modern recent literature on quadruple systems. This practice may bias the context and undermines the breadth of scholarship expected in a high-impact journal.

The manuscript presents a valuable discovery and a thorough observational analysis. However, it requires major revisions to address the following:

- Re-evaluate the evolutionary scenario for the inner binary, considering stable, conservative mass transfer as the likely outcome.
- Include dynamical stability analysis using direct or secular N-body simulations.
- Reduce self-citation and broaden the literature context.
- Remove irrelevant content (e.g., X-ray binaries) and condense the manuscript.
- Refocus the discussion of formation mechanisms to those directly relevant to the system.
- Moderate the use of subjective and superlative language.
- Emphasize the well-constrained aspects of the system's past and formation over speculative future evolution.
- Support the argument for the common-envelope evolution in a radially stable non-Darwin-Riemann unstable system.

Addressing these issues will dramatically improve the manuscript's scientific rigor, relevance, and clarity.

Reviewer #2

(Remarks to the Author)

This manuscript describes an incredibly unique, well characterized quadruple system. The very rare hierarchy: a compact triple with a wider 4th object, is both rare and hard to explain theoretically. The well known masses, radii, and orbit configuration in this system make it worthy of further study. The analysis of the likely evolutionary tracks of the system with MESA are a particularly nice complement to the discovery itself.

The manuscript itself clearly explains the analysis process undertaken to obtain the full orbit fit and derive stellar parameters. The orbital fits and stellar parameter derivation appear sound and are based on well vetted procedures used commonly in the field. My main comments are stylistic. As written, the manuscript presents more of a "story" of the discovery of the object, going through the original, incorrect, orbit solutions, and describing how the authors uncovered the 4th object. I would

strongly recommend that the authors present the result in a narrative less driven by the order of discovery, and more driven by the result. For example, it is important to explain why the three star fit is insufficient. It is not necessary, in my opinion, to present it in chronological order. In particular, I think that re-imagining the narrative structure in this way might lead to a much more concise presentation.

The second main stylistic recommendation I have would be to organize the information about the different data sources into a table of some kind (separate from the actual observational data tables presented in the appendix). The beginning of section 2 is very hard to follow with the lists of different observatories and numbers of nights of data. It would be more helpful for the reader to summarize which data came from which observatory, and perhaps a column that summarizes its role in the analysis (for example making clear that the TRES data revealed the 4th spectrum).

Finally, I have a minor comment on the discussion of the MESA results. While the authors make well motivated assumptions about uncertain parameters (convective overshoot, conservative mass transfer), they do not explain or explore how these choices might qualitatively impact their final results. While a full parameter study is beyond the scope of this paper, it would be beneficial for the authors to at least comment on the ways in which the final outcome might depend on these choices.

Reviewer #3

(Remarks to the Author)

In this paper, the authors announce the discovery of TIC 120362137, the currently most compact known 3+1 (or 2+1+1) hierarchical quadruple star system. Compact multiple-star systems are highly interesting for testing dynamical and stellar evolution models of multi-stellar systems, and discovering and accurately classifying more of such systems is a valuable contribution to the field, especially given the so-far very small number of such known systems. Especially significant are some of the stellar evolution model predictions the authors produce for this system, since it appears that this quadruple system will eventually become a (compact) double white dwarf system, completely erasing the current dynamical configuration for future observers. This can have interesting implications for the evolutionary history of the known population of double white dwarf systems, such as those observed by Gaia.

In their work, the authors competently outline the discovery of the system from a combination of space- and ground-based observations, as well as provide a detailed analysis of the data using a variety of established analysis programs. I cannot see any significant weaknesses in the author's analysis of the data and their conclusions seem sound. I have, however, a number of suggestions to improve various elements of the paper in terms of clarity, which I will outline here:

1. Starting at Line 200, the authors mention that some data was taken at private observatories in the Czech Republic. Some more information on these observatories would be prudent to add. In general, some of the observatories mentioned in this section are not referenced - links to any public websites or references to documents outlining the technical details of these observatories and their instruments could be helpful.

2. In Line 259, the phrasing of "[...] we were able to constrain the physical and dynamical parameters of the hypothetical fourth body to somewhat realistic values." is rather vague - instead of saying "somewhat realistic", it would be better to qualify in what way the parameters are realistic (i.e. orbital stability, characteristics of the lightcurves, etc.).

3. Fig. 1: The plots are too small and very hard to read. The data is also given without error bars - even when using binned data, it would be helpful to have an indication of the typical size of the errors. My suggestion would be to have one larger example frame that shows the unbinned data including the error bars, or at least one example data point with an error bar.

4. Fig. 2: The plots are again entirely too small. The mentioned gray data points are basically not visible, a different choice of colors and marker symbols would be prudent.

5. Fig. 3: These plots are very well crafted and very insightful - however, again, at least one example of the typical size of the error bars would be advisable.

6. Fig. 4: Error in the caption: The blue symbols are squares, not circles. Furthermore, the "thick" vs "thin" lines for the TESS vs ground-based observations of the eclipses are not distinguishable. I would advise using a dashed line for one of them, instead of varying the thickness.

7. Fig. 5: The gray line for the COM orbit is not visible. Another, more visible color choice would be good - the orbits of the inner two objects could be plotted in more transparent colors, with the COM orbit colored more opaquely to stand out.

8.: Line 360: The flatness of the quadruple system could also be argued to be necessary for long-term dynamical stability, thus resulting in only primordially aligned systems surviving to be observed. There are various ways to test the stability of such systems, beyond conducting n-body simulations, which the authors may want to review - the Naoz 2016 paper the authors cite lists a number of stability criteria, such as the condition outlined by Mardling & Aarseth (2001), which does take inclination into account. The authors could use such a criterion to estimate a maximum possible mutual inclination for the outer orbits. In general, a more detailed discussion of the long-term stability of compact multi-star systems could be helpful.

9.: Line 480: The authors have structured the paper sections oddly, in my opinion, and are only listing their methods here, after already discussing all data and results. I personally would list the methods before the results (since the analysis methods are needed to produce the results), after the data/observations section; however, if this is a particular structure style

that is preferred by Nature Communications, I will retract my comment.

10: Note on significant figures - at various parts of the paper, such as in the conclusions, the results are listed with a different degree of accuracy than as they are listed in the result tables. I believe it would be better to be consistent one way or another.

Once the authors have resolved these listed issues, I believe that their paper is ready for publication.

Reviewer #4

(Remarks to the Author)

The authors present a first comprehensive study of the rare quadruple hierarchical (3+1) stellar system. Observational data have been collected using various techniques, photometry, spectroscopy and ETV. Data from the TESS space telescope were used in the analysis. The system shows eclipses of the three inner components. A scenario for the future evolution of the system is presented. The final stage of evolution according to the authors is a double white dwarf system. The authors carried out numerical simulations of the motion of the components of the system. The object is the most compact 3+1 known system. The authors mention that there are two similar systems, however TIC 120362137 is the only one for which spectroscopic detection of all four components has been successful, in addition to being the tightest of the systems. Multiple systems and their evolutionary scenarios are still very important from the point of view of star formation processes.

The technical side of the paper looks very good, the text is understandable and the plots are clear.

The authors carried out a detailed careful analysis of this rare system. The work is worthy of publication, I have only some minor comments for the authors.

MAJOR ISSUES

I have three comments that are not necessary to include however are worthy of consideration by the authors.

1. Can You show exemplary plot of cross-correlation results, showing peaks for all four components (BF or/and CCF, end of section 2)? Such plots are very informative.
2. In the paper, You are showing only osculating elements. Can You show the estimate, in what range do semi-major axes and eccentricities change due to the gravitational perturbations from components in whole system?
3. Regarding perturbations, I assume that they do not lead to destruction of the system within 275 mln years of evolution presented in chapter 3.2 and on Fig. 6, but providing such statement in the text should clarify the situation.

MINOR ISSUES

1. In the Fig. 3, upper right and both lower plots have subscription "BJD - 2400000", when in my opinion they present phase on X-axes. Please correct that.
2. Inconsistency in P3 value: in abstract, and Fig. 6 and 7 the value is $P_3=1045$, when in lines 441 (chapter 3.3) and 645 (chapter 5) the value is $P_3=1046$. As in Tab. 1 provides value $P_3=1045.5$.
3. In line 678 You stated, that final period of WD-WD system will be ~ 51.4 day, but in Fig. 7 the value is ~ 44 days - which value is correct?
4. Line 223, there seems to be a space missing: 7250Awawelength

Reviewer #5

(Remarks to the Author)

The manuscript, titled "Discovery of the most compact 3+1-type quadruple system, TIC 120362137", reports a compact quadruple star system that can fit between the Sun and the orbit of Jupiter. The strength of the manuscript is the advanced setup to simultaneously model the triply eclipsing light curves, eclipse timing variations, as well as radial velocities (obtained from all four stars). This work is commendable for its computational and observational planning. The work also shows that such systems can be progenitors to wide white-dwarf binaries. Such a system challenges current ideas of star formation and evolution as well. I recommend the manuscript for publication, but with some corrections:

-Section 2 (important): The authors present an interesting system where all 4 stars are visible in the spectra. Therefore, it would be useful to have figures for CCF/BF/spectra for at least two epochs with all the components visible and labelled.

-Section 3.1: The precision of physical parameters is very impressive. But what interests me is the precision on temperatures of Aa, Ab, and B. The precision is almost at the level of interferometric temperature estimates (Karovicova et al., 2018, MNRAS, 475, L81). Is this because of additional constraints from eclipse depth ratios or a result of the constraint from isochrones? If it is a result of the isochrone constraints, how much will a change in the isochrone model impact the errors on

the temperature change?

-Section 3.1: Is it possible that the stars in such a compact system interacted before? If yes, can such a case affect the parameter estimate of the fourth star?

- Section 3.2: Do you include non-synchronous rotation (as mentioned in section 2) in the evolution modelling? Will it impact the evolution and dynamics compared to a synchronous model?

-Section 3.2 (important): The authors mention that "if such a double WD system is found today, the observers would likely have no idea that it might have come from such an exotic compact 3+1 quadruple system". The mass for one of the expected WD is less than 0.3 solar masses. Such a low-mass WD can only be formed via an interaction scenario. This could also be checked by comparing TIC 120362137's final stage to available WD-WD binary samples (e.g., El-Badry et al., 2021, MNRAS, 506, 2269). If possible, the expected mass and age of the final WD binary could be compared to the distributions available in Heintz et al., 2022, ApJ, 934, 148.

Reviewer #6

(Remarks to the Author)

Version 1:

Reviewer comments:

Reviewer #1

(Remarks to the Author)

To the Authors,

Review of the Manuscript: Discovery of the most compact 3+1-type quadruple star system, TIC 120362137 by Tamas Borkovits et al., submitted to Nature Communications.

First, please accept my apologies for the delay in this report. Summer brought a range of demanding activities needing attention. At the same time, I thank you for your efforts in improving the manuscript. In fact, I think that the paper has improved considerably.

As noted in previous reports, the manuscript presents the discovery and detailed observational analysis of TIC 120362137, proposed as the most compact known 3+1 hierarchical quadruple star system. The authors effectively combine photometric and spectroscopic data to characterize the system and discuss its evolutionary history.

In this revised version, the authors have convincingly demonstrated their key findings. However, the dynamical analysis and stellar evolution sections raise concerns. Using direct IAS integration to model the stable four-body system is inappropriate as numerical errors bias stability results. I recommend removing the section on numerical stability.

Similarly, applying MESA to simulate the evolution of multi-star systems is unsuitable, as it is designed primarily for single star evolution. Ideally, coupled secular dynamical and stellar evolution models should be employed, though I acknowledge the complexity involved. Treating dynamical and stellar evolution separately underrepresents this system's complexity.

Regarding the binary evolution, the statements on common envelope evolution and Roche lobe overflow are speculative. The system is likely dynamically stable, and the binary's mass transfer stability should be reassessed. I suggest removing the binary (quadruple) evolution section and Figure 6.

The manuscript remains longer than necessary. Please focus on the

observational data and key conclusions, leaving deeper theoretical interpretation to future work. The writing could still be polished to reduce colloquial expressions.

It is commendable that the number of self-citations has been reduced, reflecting the first author's contributions more fairly. Thank you for addressing this.

To summarize, previous recommendations mostly stand:

Reconsider the evolutionary scenario for the inner binary; current treatment is inadequate and should be removed.

Replace the stability analysis based on direct integration with an appropriate secular dynamical analysis or remove it.

Focus the manuscript more tightly on observations, reducing length and speculative content.

Moderate subjective language and emphasize well-supported conclusions.

Focus discussion on well-constrained system history rather than uncertain future predictions.

Address the argument for common-envelope evolution in a radially stable non-Darwin unstable system, which remains unaddressed.

As a final remark and recommendation. It might be better to focus on the discovery, and that excitement, rather than putting any emphasis on the dynamical and stellar evolution of the system. It turns out too much of a challenge to say anything that makes sense at this point. Adopt Aarseth-Marding and the system's age to argue it's dynamical stability, and report on the primary being the first to leave the main-sequence and commencing in RLOF. Leave it by that, and do not try to predict the system's future by unreliable stellar evolution calculations or dynamical simulations.

Thank you for your efforts.

Reviewer #2

(Remarks to the Author)

The authors revisions address all of my concerns with the manuscript, and I would recommend it for publication.

Reviewer #3

(Remarks to the Author)

The authors have extensively reworked their manuscript, incorporating and responding to the comments of myself and the other reviewers. In my view, the paper is in an overall good state and I do not have any significant additional comments, and I recommend the paper to be published.

Reviewer #4

(Remarks to the Author)

I have accepted the revised text

Reviewer #5

(Remarks to the Author)

The authors have made substantial revisions to the manuscript in response to the questions posed by all reviewers. I find the answers to my question satisfactory and recommend the manuscript for publication.

Reviewer #6

(Remarks to the Author)

I co-reviewed this manuscript with one of the reviewers who provided the listed reports. This is part of the Nature Communications initiative to facilitate training in peer review and to provide appropriate recognition for Early Career

Researchers who co-review manuscripts.

Version 2:

Reviewer comments:

Reviewer #1

(Remarks to the Author)

Dear Authors,

I apologies for the enormous amount of this this all takes, but I did read your paper again, and your response to my earlier comments. In the end, I think that the paper has improved considerably. I do appreciate your responses to my rather critical comments, but I do see you point, and accept your explanations.

Reviewer #3

(Remarks to the Author)

The paper is in a good state for publication, and I accept it without needing any further revisions.

Reviewer #1 (Remarks to the Author):

To the authors:

Review of the Manuscript: Discovery of the most compact 3+1-type quadruple star system, TIC 120362137 by Tamas Borkovits et al, submitted to Nature Communications.

The manuscript reports the discovery and detailed observational analysis of TIC 120362137, claimed to be the most compact known 3+1 hierarchical quadruple star system. The authors combine photometric and spectroscopic data to characterize the system and discuss its evolutionary history and future.

We thank the Reviewer for his/her very detailed comments. We agree with the Reviewer on the majority of his/her comments. However, there are a few points, where we have a different perspective and, therefore, we have slightly different opinions. We answer point by point all the Reviewer's statements. All the modifications (additions) to the manuscript are denoted with magenta characters. Note, moreover that, in accord with the requirements of the Editor, we restructured the entire paper, following the style guidance of Nature Communications.

Despite the manuscript's claim of being a "dynamical study," no direct N-body or secular N-body simulations are presented. The authors

do not address the dynamical stability of the system, which is crucial for such a compact and hierarchical quadruple. Without such analysis, the long-term viability and evolutionary predictions for the system remain speculative and incomplete. In particular, such a study is also important for the \sim Gyr stability of the system.

There is a slight misunderstanding about the main purpose of the current paper. We do not claim that this is principally a dynamical study! Instead, this is an observational discovery with an extensive analysis of the system properties. We guess that the term, '[spectro-]photodynamical analysis' was what led the Reviewer to such a misunderstanding. The adjective 'photodynamical' in our narrower field of astrophysics means simply such an analysis of observational data of multiple stellar (and/or) planetary systems, where departures from purely Keplerian motions, due to direct third- and fourth-body perturbations cannot be neglected, despite the short time-window of the observations. Therefore, a numerical integration of the motion, included in the analysis, is necessary. This approach, as far as we know, was first used in connection with the Kepler-spacecraft observations KOI 126 by Carter et al., 2011, Science, 331, 562. Since then, [spectro-]photodynamical analyses are in somewhat widespread use in fitting data from transiting exoplanetary systems, as well as multiply eclipsing stellar systems. In any case, we are really very thankful to the Reviewer for pointing out the possibility of a misunderstanding. Therefore, now we explain in the text (in the next to last paragraph of the Introduction) what we mean by the term of "photodynamical analysis" in order to avoid this kind of

misunderstanding. Independent of this, the critique of the Reviewer regarding the lack of any long-term numerical integration(s) regarding the long-term stability of the system is absolutely correct. Now we have carried out this kind of integrations, and the results are introduced and discussed in Section 3.

The introduction discusses the evolution of X-ray binaries, which is not relevant to the subject of this paper. This distracts from the focus and dilutes the impact of the manuscript.

We substantially shortened the entire paper and, therefore, the Introduction as well. As a part of this shortening, we deleted any mention of X-ray binaries, and note simply that the (final) evolution of multiple star systems might lead to various exotic objects and end states.

The paper is unnecessarily long, with significant portions not directly relevant to the main discovery or analysis. This detracts from readability and clarity, and the manuscript would benefit from substantial condensation.

The new revised version, which follows the formatting requirements of Nature Communications, naturally is shorter, by at least 30-40%. We made serious efforts to generate a more concise version.

The discussion on the formation mechanism is not clearly applicable or relevant to the specific system under study. The formation

scenario

should be directly tied to the observed properties and uniqueness of

TIC 120362137, rather than relying on generic or loosely connected arguments.

Our purpose is not to speculate on the formation mechanism(s) of such a system. According to our knowledge, there are three different kinds of multiple star formation processes: i) core or filament fragmentation, (ii) (multiple) disk fragmentation, and (iii) capture. We are not really experts on this topic, but as far as our understanding is correct, amongst these mechanisms only number (ii) might produce such a compact, coplanar quadruple. Here, what we do is simply stress this point, and give citations for some review papers which naturally give further, more detailed references. Carrying out extensive hydrodynamical simulations to check whether such mechanisms might result in such a compact quadruple is, however, at least in our understanding, far outside the scope of the current paper.

to this extent, the term "To the best of our knowledge" does not give

sufficient warrant for original research. The authors should do their homework, and make sure that it either "is" or "is not" the most

compact quadruple discovered so far. The point itself, however, is rather irrelevant. Maybe, it would be better if the authors just ignore the entire mention of "first" and "best", and state the facts, rather than their desire for novelty.

The Reviewer is correct that our expression "To the best of our knowledge" was not very precise and, even vague. Now we have corrected that with a more detailed discussion, as follows: "We consulted with the newest version of the Multiple Star Catalog \citep[MSC][]{2018ApJS..235....6T} and also carried out an extensive literature search, but according to these sources,..."

On the other hand, we think that it is not an irrelevant question as to whether this is the most compact quadruple stellar system ever discovered, or not. From our perspective, it is exactly the uniqueness of this quadruple system (or, in the word's of the Reviewer: 'its novelty'), which makes this work worthy of publication in such a highly ranked journal as, e. g. Nature Communications. In our view, for example, the existence of, and several billion years of survival for, such an extremely compact quadruple system might offer very strong constraints on multiple star formation theories, similar to, e.g., the clearly very important discovery of the very first quadruple proto-star system (which, however, had a characteristic size that is ~ 3 orders of magnitude larger), where the discovery was published ten years ago in Nature (Pineda et al. 2015, Nature, 518, 213).

The authors predict that the inner binary will undergo a common envelope (CE) phase upon the primary filling its Roche lobe, leading to a merger. However, this scenario is questionable. For such binaries, the first mass transfer episode is more likely to be stable and largely conservative, rather than leading to a CE phase. The mass transfer would cause the inner orbit to expand, as well as the orbits

of the outer companions. Only a small fraction of the mass is typically lost from the system in these cases. This omission leads to an inaccurate depiction of the likely evolutionary pathway.

By the time that mass transfer from Aa to Ab commences, the primary star has a radius of 5.44 R_{sun} , and has developed a convective outer structure. Because the donor star is more massive than the accretor, the orbit shrinks in response to the mass transfer and the donor star is insufficiently radiative to contract in response to the mass loss. Therefore, stable thermal timescale mass transfer does not occur. Our detailed MESA calculations show that the mass transfer rates exceed 10^{-4} and even 10^{-3} $M_{\text{sun}}/\text{year}$, the Roche lobe of the accretor becomes filled, and the MESA calculations stop. In these calculations we have tested mass loss fractions from 0 to 70%, and have considered the specific angular momentum carried away by any ejected matter to be that of the accretor. If we were to assume that the specific angular momentum of the ejected matter were that of the L2 point in the system, the mass transfer event would be even far more unstable. These are all solid indicators that the mass transfer is indeed unstable as we have stated in the paper. This is consistent with previous studies, such as Chen & Han 2008 (<https://ui.adsabs.harvard.edu/abs/2008MNRAS.387.1416C/abstract>), which gives the critical mass ratio for dynamical mass transfer (see their table 1). If the referee can point us to any published articles that include a case with roughly comparable system parameters, where the mass transfer is shown to be stable, then we will reconsider our conclusion in this regard.

We continue by assuming that the stars merge via a common envelope, and ask the question about what fraction of the mass

has been lost from the system after the final merger. Since any of the prescriptions for ejecting a common envelope, we find that, with the alpha parameter set as high as unity, there is by far insufficient gravitational binding energy to be released to eject any significant fraction of the donor star's envelope. Even if we set alpha equal to 3 or 10 (for which there is no justification, because there is insufficient energy available in recombination of ions in the envelope), there is still insufficient energy to eject a substantial portion of the common envelope.

Thus, the net result of the first mass transfer event is a nearly complete stellar merger following a common envelope, with very little mass loss.

This discussion of the system's future, being highly uncertain, and depends critically on dynamical stability and the stability of mass transfer. In particular the latter is one of the least certain aspects of binary evolution. The manuscript would be stronger if it focused more on the past and formation of the system, which are more constrained by the data, rather than speculative future scenarios. How do one get four similarly-massive stars with very small mutual inclination in almost circular orbits.

We agree with the Reviewer that the discussion of the system's future is somewhat speculative. However, we believe that the future evolution can be more robustly and even quantitatively evaluated than the system's formation history. We have carried out simulations, starting from very robustly known current set of system parameters, simulating the future of the system with the use of a software package that is in widespread use and is considered to be state-of-the-art. Of course, we necessarily must

adopt some assumptions for some parameters that are not calculated directly from MESA, such as the specific angular momentum carried away by ejected matter. However, we feel that we have done a credible job that results in some things that are worth pondering.

The use of a common stellar evolution code is no guarantee here. Common binary evolution theory will render this system stable against common envelope. The authors should at least provide a thorough analysis of the Darwin-Riemann (or other) instability criteria that initiates the CE.

For the inner EB, assuming the rotation velocity is at the break-up velocity, we can find the rotational angular momentum. In this case, we can estimate the ratio of rotational angular momentum to the orbital angular momentum, which can not be larger than 0.20. Therefore the Darwin-Riemann instability will not occur. Furthermore, the Darwin instability takes place on the tidal dissipation timescale which is what ultimately causes the orbit to decay. On the other hand, once the mass transfer actually starts, and gets up to 10^{-3} Msun per year, the mass transfer becomes unstable on a dynamical timescale. So, it is nearly irrelevant whether there is or is not a Darwin instability operating. In other words, just stellar evolution plus unstable mass transfer will cause the system to merge.

Please, check for orbital resonance, in particular check to what extent the quadruple is in mean-motion resonance.

In hierarchical configurations, like the current quadruple system, mean-motion resonances have lesser significance than in planetary systems. From a mathematical point of view, this is due to the larger $P_{\text{out}}/P_{\text{in}}$ (and hence, $n_{\text{in}}/n_{\text{out}}$) ratio(s). In this current quadruple, for example, $P_{\text{mid}}/P_{\text{in}}=15.6$ and, $P_{\text{out}}/P_{\text{mid}}=20.4$. Consequently, such ratios certainly cannot be approximated with the ratios of small integers. Therefore, even if we would find some kind of mean-motion “resonance” it would be of a very large order and, hence, very small amplitude for any dynamical significance. As an example, here we mention the case of the compact triply eclipsing triple star, HD 181068 where the period ratio of the circular inner and outer orbits has an almost exact value of $P_{\text{in}}/P_{\text{out}}=5:251$ (representing the true period ratio to a fractional accuracy of 0.1%). This nicely defines five different families of third-body light curves or, outer eclipses – that is, every fifth primary or secondary third-body eclipse looks very similar due to the very similar spatial configurations of all three stars. But, this has no effect on the dynamics of that triple – see Borkovits et al., 2013, MNRAS, 428, 1656. Of course, we cannot exclude the possibilities of any secular spin-orbit resonances in the past, present or, future of the current quadruple (see, e.g., Correia et al., 2016, CeMDA, 126, 189), but such investigations, in our opinion, are beyond the scope of the current paper.

Terms such as "significant," "the most compact" (used 27 times, including in the title), and "for the first time" are overused. Such language should be reserved for clear, quantifiable distinctions and not repeated excessively, as it can undermine scientific objectivity.

We have made efforts to reduce the number of occurrences of such superlatives throughout the text. On the other hand, as we discussed above, we feel that it is important that the current quadruple is the most compact known 3+1 type system and, hence, we kept the expression “most compact” in some places. Moreover, we note, however, that according to our search, the expression “for the first time” was used only once in the whole paper. Even when we searched simply for “for the first” we found one additional phrase in the entire text. Therefore, while we reduced the numbers of such expressions as “significant” and “most compact”, we left some of them in the text.

The first author cites his own work 20 times out of ~60 references. This is excessive and inappropriate, especially given that the author does not dominate the mondial recent literature on quadruple systems. This practice may bias the context and undermines the breadth of scholarship expected in a high-impact journal.

It is true that none of the authors dominate the literature of quadruple stellar systems in general. However, we feel that in the subfield of the observational analysis of compact multiply eclipsing stellar systems, the situation is quite different. And, perhaps, the current authors might be a little proud of the fact that in the past few years they have discovered and analysed the currently shortest known outer period 2+1 type triple (24.5 d), as well as the 4-5 shortest outer period 2+2 type quadruple stellar systems (121-168 d), not to mention the first two (2+2)+2 type sextuple stellar systems, which contain three eclipsing binaries...

Moreover, we claim that the primary purpose of this paper is to present the discovery (and, of course, analysis) of such an unusually compact, 3+1 type quadruple system which was obtained from the analysis of an object which was originally thought to be simply a “usual” triply eclipsing triple stellar system. Despite all this, we agree with the Reviewer, that the presence of so many self-citations is a bit too self-serving. Therefore, we revised all these citations, deleted many, and kept only those which we felt were necessary. Therefore, we removed 8 self citations. (In half of them the current first author was the first author there as well). Moreover, we added some new, non-self citations at a number of places. Currently the paper contains 66 citations, from which the current first author was first author in 6 papers (less than 10%). Moreover, there are 10 additional papers which have at least one common author with the current manuscript, however, some of these papers refer only to some technical details on the instruments and methods which were used and, therefore, they cannot be removed. We hope, that these numbers are now more acceptable for the Reviewer.

The manuscript presents a valuable discovery and a thorough observational analysis. However, it requires major revisions to address the following:

- Re-evaluate the evolutionary scenario for the inner binary, considering stable, conservative mass transfer as the likely outcome.

We believe we have addressed this issue at length above, and we simply respectfully disagree with the referee about the stability of the ensuing mass transfer in the Aa-Ab system.

- Include dynamical stability analysis using direct or secular N-body simulations.

These have now been included.

- Reduce self-citation and broaden the literature context.

Done.

- Remove irrelevant content (e.g., X-ray binaries) and condense the manuscript.

We discussed this above, removed some irrelevant content, and made efforts to condense the manuscript substantially.

- Refocus the discussion of formation mechanisms to those directly relevant to the system.

As we explained above, we do not feel the necessity for discussing of any formation mechanism(s) in a more detailed manner.

- Moderate the use of subjective and superlative language.

Done.

- Emphasize the well-constrained aspects of the system's past and formation over speculative future evolution.

This has been discussed above. e still believe that the future evolution is on a more secure and quantitative footing

than the formation scenarios.

- Support the argument for the common-envelope evolution in a radially stable non-Darwin-Riemann unstable system.

This has also been discussed above.

Addressing these issues will dramatically improve the manuscript's scientific rigor, relevance, and clarity.

We thank the Reviewer again for the helpful recommendations. We hope that, despite the fact that we did not agree with the Reviewer on all the points, his/her extensive critiques and suggestions have helped to substantially improve the scientific rigor, relevance and clarity of this manuscript.

Reviewer #2 (Remarks to the Author):

This manuscript describes an incredibly unique, well characterized quadruple system. The very rare hierarchy: a compact triple with a wider 4th object, is both rare and hard to explain theoretically. The well known masses, radii, and orbit configuration in this system make it worthy of further study. The analysis of the likely evolutionary tracks of the system with MESA are a particularly nice complement to the discovery itself.

We thank the Reviewer for his/her comments. We answer point by point all the Reviewer's statements. All the modifications (additions) to the manuscript are denoted with magenta characters. Note, moreover that, in accord with the requirements of the Editor, we restructured the entire paper, following the style guidance of Nature Communications.

The manuscript itself clearly explains the analysis process undertaken to obtain the full orbit fit and derive stellar parameters. The orbital fits and stellar parameter derivation appear sound and are based on well vetted procedures used commonly in the field. My main comments are stylistic. As written, the manuscript presents more of a „story” of the discovery of the object, going through the original, incorrect, orbit solutions, and describing how the authors uncovered the 4th object. I would strongly recommend that the authors present the result in a narrative less driven by the order of discovery, and more driven by the result. For example, it is important to explain why the three star fit is insufficient. It is not necessary, in my opinion, to present it in chronological order. In particular, I think that re-imagining the

narrative structure in this way might lead to a much more concise presentation.

We reformulated the description of how our observations led to the discovery of the fourth component, and also tried to make it clear in the new Sect. 2.1 why a three-body solution is insufficient.

The second main stylistic recommendation I have would be to organize the information about the different data sources into a table of some kind (separate from the actual observational data tables presented in the appendix). The beginning of section 2 is very hard to follow with the lists of different observatories and numbers of nights of data. It would be more helpful for the reader to summarize which data came from which observatory, and perhaps a column that summarizes it's role in the analysis (for example making clear that the TRES data revealed the 4th spectrum).

We have added such a table with the log of the observations to the appendix (which, according to our suggestion to the Editor, might form part of the Supplementary Material – at least, in the case where our paper has been accepted); however, we did not include an extra column with the role that the current data plays in the analysis. The purpose of each data set, being photometric or spectroscopic, more or less tells its purpose.

Finally, I have a minor comment on the discussion of the MESA results. While the authors make well motivated assumptions about uncertain parameters (convective overshoot, conservative

mass transfer), they do not explain or explore how these choices might qualitatively impact their final results. While a full parameter study is beyond the scope of this paper, it would be beneficial for the authors to at least comment on the ways in which the final outcome might depend on these choices.

With reasonable assumptions about convective overshooting, we find that the impact of overshooting has only a minor effect on the evolution and the overall evolution scenario will not change. How conservative the mass transfer is, in particular during the merging process, will influence of the masses of stars A' and AB. This will also influence the WD mass and orbital period in the final double WD binary. But the overall evolution scenario should be very similar to the description in Fig. 6. We have added some discussion about these points in Sec. 3.

Reviewer #3 (Remarks to the Author):

In this paper, the authors announce the discovery of TIC 120362137, the currently most compact known $3+1$ (or $2+1+1$) hierarchical quadruple star system. Compact multiple-star systems are highly interesting for testing dynamical and stellar evolution models of multi-stellar systems, and discovering and accurately classifying more of such systems is a valuable contribution to the field, especially given the so-far very small number of such known systems. Especially significant are some of the stellar evolution model predictions the authors produce for this system, since it appears that this quadruple system will eventually become a (compact) double white dwarf system, completely erasing the current dynamical configuration for future observers. This can have interesting implications for the evolutionary history of the known population of double white dwarf systems, such as those observed by Gaia.

We thank the Reviewer for his/her comments. We answer point by point all the Reviewer's statements. All the modifications (additions) to the manuscript are denoted with magenta characters. Note, moreover that, in accord with the requirements of the Editor, we restructured the entire paper, following the style guidance of Nature Communications.

In their work, the authors competently outline the discovery of the system from a combination of space- and ground-based observations, as well as provide a detailed analysis of the data using a variety of established analysis programs. I cannot see any significant weaknesses in the author's analysis of the data and

their conclusions seem sound. I have, however, a number of suggestions to improve various elements of the paper in terms of clarity, which I will outline here:

1. Starting at Line 200, the authors mention that some data was taken at private observatories in the Czech Republic. Some more information on these observatories would be prudent to add. In general, some of the observatories mentioned in this section are not referenced - links to any public websites or references to documents outlining the technical details of these observatories and their instruments could be helpful.

In accordance with the recommendations of another Reviewer, we have now introduced a new table in the Appendix of the paper with the log of the ground-based observations, and appropriate references. We hope the Table makes this part of the paper more informative. Moreover, we added some additional references (papers as well as URLs) for some of the Czech Observatories.

2. In Line 259, the phrasing of "[...] we were able to constrain the physical and dynamical parameters of the hypothetical fourth body to somewhat realistic values." is rather vague - instead of saying "somewhat realistic", it would be better to qualify in what way the parameters are realistic (i.e. orbital stability, characteristics of the lightcurves, etc.).

This question, in accordance with the suggestions of other reviewers, is now discussed in later parts of the paper. Therefore, at this point in the paper, we simply mention that this question will be discussed later.

3. Fig. 1: The plots are too small and very hard to read. The data is also given without error bars - even when using binned data, it would be helpful to have an indication of the typical size of the errors. My suggestion would be to have one larger example frame that shows the unbinned data including the error bars, or at least one example data point with an error bar.

Now, we have replaced these figures with two representative plots (as Fig. 1). One of them displays a large, single-dip third-body eclipse, while the second one (the right panel) shows a two-dip third-body eclipse event. We added a residual flux curve below each panel, and also show typical error bars. We plan to present the bulk of the third-body plots in some on-line only repositories. Note, we put the nine panels of the original Fig. 1 into the appendix, and have suggested to the Editor that, in the case that the paper is accepted, it might form the part of the Supplementary Material.

4. Fig. 2: The plots are again entirely too small. The mentioned gray data points are basically not visible, a different choice of colors and marker symbols would be prudent.

We moved these figures into the appendix, judging them to be less informative. However, we suggest that, together with all the original panels of Fig.~1, they might form part of the Supplementary Material.

5. Fig. 3: These plots are very well crafted and very insightful - however, again, at least one example of the typical size of the error bars would be advisable.

Thank the reviewer for the comment about the quality of these figures. We note that the error-bars are plotted in the lower (residual) panels of each plot. Now we call the attention of the Reader to these error bars.

6. Fig. 4: Error in the caption: The blue symbols are squares, not circles. Furthermore, the "thick" vs "thin" lines for the TESS vs ground-based observations of the eclipses are not distinguishable. I would advise using a dashed line for one of them, instead of varying the thickness.

Thank you for calling our attention to this error. We have corrected them. Moreover, we changed the "thin" lines to dashed ones.

7. Fig. 5: The gray line for the COM orbit is not visible. Another, more visible color choice would be good - the orbits of the inner two objects could be plotted in more transparent colors, with the COM orbit colored more opaquely to stand out.

Now we have plotted the COM orbit in brown color, and overplotted the Aa's and Ab's orbit with that one. We feel that the COM orbit is now somehow more visible. We note that we have also tried different color combinations. In some other versions, the COM orbit is more visible, but we found those other versions to be aesthetically not so good.

8.: Line 360: The flatness of the quadruple system could also be argued to be necessary for long-term dynamical stability, thus resulting in only primordially aligned systems surviving to be

observed. There are various ways to test the stability of such systems, beyond conducting n-body simulations, which the authors may want to review - the Naoz 2016 paper the authors cite lists a number of stability criteria, such as the condition outlined by Mardling & Aarseth (2001), which does take inclination into account. The authors could use such a criterion to estimate a maximum possible mutual inclination for the outer orbits. In general, a more detailed discussion of the long-term stability of compact multi-star systems could be helpful.

Yes, it was a weakness of the paper that in the original version (primarily to save some spaces) we left out any stability studies. Now we have included such studies, using both the slightly modified form of the Mardling & Aarseth (2001) condition, and also using detailed 4-body integrations (with the use of two different numerical integrators).

9.: Line 480: The authors have structured the paper sections oddly, in my opinion, and are only listing their methods here, after already discussing all data and results. I personally would list the methods before the results (since the analysis methods are needed to produce the results), after the data/observations section; however, if this is a particular structure style that is preferred by Nature Communications, I will retract my comment.

The final structure of the paper differs from the original version, and follows the style requirements of Nature Communications.

10: Note on significant figures - at various parts of the paper, such as in the conclusions, the results are listed with a different degree

of accuracy than as they are listed in the result tables. I believe it would be better to be consistent one way or another.

The reason for the different numbers is as follows: In the tabulated results, we typically give one digit more than is significant. And, moreover, we give formal upper and lower 1-sigma uncertainties. These are to be considered the definitive results with all the appropriate significant figures and realistic uncertainties. In contrast to this, in the text itself, in particular in the “Current physical and dynamical parameters” section we give only one or two decimal points (and no uncertainties) to make comparisons among the different stars easier for the reader. Thus, we don’t feel that there should be any confusion stemming from the values given. We note, however, that if the Reviewer does not agree with our decision, and asks us to give exactly the same values and uncertainties in the text as in Tables 1-2, we would gladly fulfill such a requirement in a next round.

Once the authors have resolved these listed issues, I believe that their paper is ready for publication.

Reviewer #4 (Remarks to the Author):

The authors present a first comprehensive study of the rare quadruple hierarchical (3+1) stellar system. Observational data have been collected using various techniques, photometry, spectroscopy and ETV. Data from the TESS space telescope were used in the analysis. The system shows eclipses of the three inner components. A scenario for the future evolution of the system is presented. The final stage of evolution according to the authors is a double white dwarf system. The authors carried out numerical simulations of the motion of the components of the system. The object is the most compact 3+1 known system. The authors mention that there are two similar systems, however TIC 120362137 is the only one for which spectroscopic detection of all four components has been successful, in addition to being the tightest of the systems. Multiple systems and their evolutionary scenarios are still very important from the point of view of star formation processes.

We thank the Reviewer for his/her comments. We answer point by point all the Reviewer's statements. All the modifications (additions) to the manuscript are denoted with magenta characters. Note, moreover that, in accord with the requirements of the Editor, we restructured the entire paper, following the style guidance of Nature Communications.

The technical side of the paper looks very good, the text is understandable and the plots are clear.

The authors carried out a detailed careful analysis of this rare system. The work is worthy of publication, I have only some minor comments for the authors.

MAJOR ISSUES

I have three comments that are not necessary to include however are worthy of consideration by the authors.

1. Can You show exemplary plot of cross-correlation results, showing peaks for all four components (BF or/and CCF, end of section 2)? Such plots are very informative.

Now we put such a plot into the paper, as (the new) Fig. 3.

2. In the paper, You are showing only osculating elements. Can You show the estimate, in what range do semi-major axes and eccentricities change due to the gravitational perturbations from components in whole system?

We have carried out numerical integrations using the extracted astrophysical and dynamical parameters of the three orbits and four stars as input parameters, in very similar manner to what was described e.g. in Borkovits et al. 2022, MNRAS 515, 3773. Note, however, that in the absence of any interesting phenomena, we do not show these variations in the new figures, but simply mention in the text that we obtained only very small, cyclic variations. These numerical integrations, however, had covered –dynamically speaking – only very short intervals. We have now carried out much longer timescale numerical integrations (using a somewhat simplified model of four point masses) with the numerical integrator REBOUND. These integrations cover an interval of 5 Myr and, as expected, do not show any signs of the disintegration of the system, at least during that time interval where the non-gravitational (e.g. stellar evolutionary effects) are negligible. This is practically the entire main-sequence lifetime of the system. (These longer-term integrations are also mentioned in the text.)

3. Regarding perturbations, I assume that they do not lead to destruction of the system within 275 mln years of evolution presented in chapter 3.2 and on Fig. 6, but providing such statement in the text should clarify the situation.

Now we have put the following clarification into the text: "At this point we should note that we assume none of the two mergers leads to the dissolution of the remaining triple, and later, binary systems. Considering the fact that such a merger, in general, is not a violent event (in contrast to a supernova-kick which may lead to catastrophic consequences for the stability of any remaining binary or multiple system) and, moreover, the remaining systems are wide enough so that such assumptions appear quite realistic."

MINOR ISSUES

1. In the Fig. 3, upper right and both lower plots have subscription "BJD - 2400000", when in my opinion they present phase on X-axes. Please correct that.

Thank you for calling our attention to these errors. These have now been corrected.

2. Inconsistency in P3 value: in abstract, and Fig. 6 and 7 the value is $P3=1045$, when in lines 441 (chapter 3.3) and 645 (chapter 5) the value is $P3=1046$. As in Tab. 1 provides value $P3=1045.5$.

Thank you for pointing out this inconsistency. In what follows, we use the value of 1046, instead of 1045 in the abstract as well.

3. In line 678 You stated, that final period of WD-WD system will be ~ 51.4 day, but in Fig. 7 the value is ~ 44 days - which value is

correct?

This was a typo and has been fixed. The correct period is 44 days. We note, however, that this typo was formerly in the Conclusion section. In the new version, after shortening and formatting the text according to the rules of Nature Communications, no Conclusion section remains. Therefore, the entire phrase with the typo has been deleted.

4. Line 223, there seems to be a space missing: 7250A
wavelength

Corrected.

Reviewer #5 (Remarks to the Author):

The manuscript, titled “Discovery of the most compact 3+1-type quadruple system, TIC 120362137”, reports a compact quadruple star system that can fit between the Sun and the orbit of Jupiter. The strength of the manuscript is the advanced setup to simultaneously model the triply eclipsing light curves, eclipse timing variations, as well as radial velocities (obtained from all four stars). This work is commendable for its computational and observational planning. The work also shows that such systems can be progenitors to wide white-dwarf binaries. Such a system challenges current ideas of star formation and evolution as well. I recommend the manuscript for publication, but with some corrections:

We thank the Reviewer for his/her comments. We answer point by point all the Reviewer’s statements. All the modifications (additions) to the manuscript are denoted with magenta characters. Note, moreover that, in accord with the requirements of the Editor, we restructured the entire paper, following the style guidance of Nature Communications.

-Section 2 (important): The authors present an interesting system where all 4 stars are visible in the spectra. Therefore, it would be useful to have figures for CCF/BF/spectra for at least two epochs with all the components visible and labelled.

Now we have put such a plot into the paper, as (the new) Fig. 3.

-Section 3.1: The precision of physical parameters is very impressive. But what interests me is the precision on temperatures of Aa, Ab, and B. The precision is almost at the

level of interferometric temperature estimates (Karovicova et al., 2018, MNRAS, 475, L81). Is this because of additional constraints from eclipse depth ratios or a result of the constraint from isochrones? If it is a result of the isochrone constraints, how much will a change in the isochrone model impact the errors on the temperature change?

This accuracy mainly comes from the fact that the light curve itself constrains the ratios of the surface brightnesses and hence, indirectly, the ratios of the temperatures. On the other hand, the absolute values of the temperatures are mainly constrained by the (cumulative) SED, instead of the isochrone models. Of course, it can be imagined that the isochrone models have some significant uncertainties in their temperatures, which we did not directly take into account. However, in our view, this results indirectly in some extra uncertainties in the age and/or in the metallicity of the system. Hence, in conclusion, the temperature uncertainties that we obtained seem to realistic for us. We mention, moreover, that in our view these are in accord with the introductory statements of Miller et al., MNRAS, 497, 2899. Now this is also discussed at the very end of the new Sect. 4.2.

-Section 3.1: Is it possible that the stars in such a compact system interacted before? If yes, can such a case affect the parameter estimate of the fourth star?

The joint analysis of the light curve, ETV curve, the SED curve and, especially the radial velocity curves do not depend on any a priori assumption about the former history of the current system. On the other hand, however, the use of any isochrones which constrains the metallicity, age, distance, passband-magnitude brightnesses of the stars, naturally, introduces some

astrophysical model-dependences into our analysis. Using the PARSEC isochrones, we assume implicitly, that there were no prior (non-gravitational) interactions, especially mass transfer, Between the stellar components. This assumption, especially for the fourth star, mainly affects only its age and the corresponding metallicity. This is so, because the orbital parameters, including its orbital period, as well as the mass of the fourth star are exclusively determined by the spectroscopic (mainly radial velocity) measurements and the photometric eclipse timings; therefore, any neglected previous interactions amongst the members of the inner triple system cannot affect the parameters we obtain for the fourth star, and its orbit. This fact was not clearly made the original text. Therefore, now we include some brief remarks about this (in the context of the entire quadruple system), together with our answer to the previous comment of the Reviewer, in the new Sect. 4.2.

- Section 3.2: Do you include non-synchronous rotation (as mentioned in section 2) in the evolution modelling? Will it impact the evolution and dynamics compared to a synchronous model?

Mentioning non-synchronous rotation in Section 2 in the originally submitted version, was a mistake, at least, in part. Now we show that the measured values of the projected equatorial rotational velocities ($v \sin i$) make it very likely that the two components of the innermost, eclipsing binary rotate synchronously, with equatorial speeds are 46 and 23 km/sec (see in the new Sect. 2.2, Lines 224-225, and compare these values with the measured ones in Sect. 2.1, Line 192), which are relatively low. This has only a very small impact on the evolution, and will lead to a slightly larger core mass and WD masses in the final double WD system.

-Section 3.2 (important): The authors mention that „if such a

double WD system is found today, the observers would likely have no idea that it might have come from such an exotic compact 3+1 quadruple system". The mass for one of the expected WD is less than 0.3 solar masses. Such a low-mass WD can only be formed via an interaction scenario. This could also be checked by comparing TIC 120362137's final stage to available WD-WD binary samples (e.g., El-Badry et al., 2021, MNRAS, 506, 2269). If possible, the expected mass and age of the final WD binary could be compared to the distributions available in Heintz et al., 2022, ApJ, 934, 148.

In our understanding, the properties of the double WD binary produced from the quadruple system are no different from those produced from an isolated binary star system. The two papers cited above deal with much wider systems than we are dealing with, where no mass transfer was ever likely to have taken place. Due to the accretion process during the formation of the double WD, the cooling age of the primary WD will be very different from that without accretion. In our simulation, we can not follow the realistic accretion process and its influence on the primary WD. So we cannot get the cooling age of the primary WD. Therefore, such a hypothetical observer could certainly infer that there had been mass transfer, but they could not tell if the original system was a binary, triple, quadruple, or higher-order multiple.

Reviewer #1 (Remarks to the Author):

To the Authors,

Review of the Manuscript: Discovery of the most compact 3+1-type quadruple star system, TIC 120362137 by Tamas Borkovits et al., submitted to Nature Communications.

First, please accept my apologies for the delay in this report.

Summer brought a range of demanding activities needing attention.

At the same time, I thank you for your efforts in improving the manuscript.

In fact, I think that the paper has improved considerably.

Thank you for your words and, moreover, for your additional efforts in reviewing our work.

As noted in previous reports, the manuscript presents the discovery and detailed observational analysis of TIC 120362137, proposed as the most compact known 3+1 hierarchical quadruple star system. The authors effectively combine photometric and spectroscopic data to characterize the system and discuss its evolutionary history.

In this revised version, the authors have convincingly demonstrated their key findings. However, the dynamical analysis and stellar evolution sections raise concerns. Using direct IAS integration to model the stable four-body system is inappropriate as numerical errors bias stability results. I recommend removing the section on numerical stability.

We agree with the Reviewer, that numerical integration of the orbits to demonstrate stability seems off the mark for two reasons. First, the orbit is manifestly and empirically stable in that it exists to a determined old age. Second, integrating for a million years when a system is 100 Myr or a Gyr is not really much of a stability test. Our main problem, however, is, that at least three Reviewers (including the current Reviewer) asked for stability tests in the revised version. And, in your previous report you said explicitly: “-Include dynamical stability analysis using direct or secular N-body simulations.” For this reason, we do not entirely understand your expectation, and what is the reason that our analysis became unacceptable to you. Moreover, the other referee, Reviewers #3 and #4, has accepted our modifications.

Of course, another possibility is to integrate the secular dynamics of the system. Such an integration, naturally, would be much faster, but it is unclear to us why the application of secular dynamics would be more acceptable and less numerically biased than a direct N-body integration of the equations of motion. After all, secular dynamics is derived after several simplifications, like doubly averaging and then truncating the perturbation function at some order and, moreover, stopping after one or a few steps of the successive

approximations, i.e., neglecting the majority of non-linear (e.g. quadruple-squared, etc.) terms. In our view, secular dynamics may be appropriate for really stable systems, and this is the current case for TIC 120362137; hence, we are convinced that it would work. But it fails to give quantitatively, or even qualitatively, correct solutions for a more or less unstable (i.e. weakly hierarchical) system. Such problems with purely secular dynamics were discussed in several recent papers, as:

<https://ui.adsabs.harvard.edu/abs/2016MNRAS.458.3060L/abstract>

<https://ui.adsabs.harvard.edu/abs/2018MNRAS.481.4602L/abstract>

<https://ui.adsabs.harvard.edu/abs/2025MNRAS.540.2422L/abstract>

and several others.) Therefore, it is unclear to us, why the integration of the secular dynamics would be more acceptable as a stability study.

Based on all these considerations, but in accord with the suggestions of the Reviewer, we decided to keep the semi-analytic stability study, stating that our system is stable in the Mardling-Aarseth sense. Moreover, we refer to the old age of the quadruple which is an empirical proof of the stability of the system. But, we deleted the entire discussion about the numerical stability study.

Similarly, applying MESA to simulate the evolution of multi-star systems is unsuitable, as it is designed primarily for single star evolution.

This is a point where, unfortunately, we do not agree with the Reviewer. While the first version of the MESA package was really developed for single stars, the effects of a companion star and, especially, mass transfer and loss from the system have been included into MESA for the past decade. Here we cite the first few sentences of the abstract of the MESA paper Paxton et al., 2015, ApJS, 220, 15:

“We substantially update the capabilities of the open-source software instrument Modules for Experiments in Stellar Astrophysics (MESA). MESA can now simultaneously evolve an interacting pair of differentially rotating stars undergoing transfer and loss of mass and angular momentum, greatly enhancing the prior ability to model binary evolution...”

Over the last decade and more the newer versions of MESA have been used to model numerous cases of binary stellar evolution, and in some cases, entire populations of binaries. For a few of many, many examples, see

<https://ui.adsabs.harvard.edu/abs/2011ApJ...732...70L/abstract>

<https://ui.adsabs.harvard.edu/abs/2016ApJ...833...83K/abstract>

<https://ui.adsabs.harvard.edu/abs/2018ApJ...854..109Z/abstract>

<https://ui.adsabs.harvard.edu/abs/2024ApJ...977..262C/abstract>

<https://ui.adsabs.harvard.edu/abs/2020ApJ...888L..12W/abstract>

<https://ui.adsabs.harvard.edu/abs/2016A%26A...588A..50M/abstract>

<https://ui.adsabs.harvard.edu/abs/2023A%26A...672A.175F/abstract>

<https://ui.adsabs.harvard.edu/abs/2021MNRAS.507.5013M/abstract>

Ideally, coupled secular dynamical and stellar evolution

models should be employed, though I acknowledge the complexity involved. Treating dynamical and stellar evolution separately underrepresents this system's complexity.

In general, we tend to agree with the referee about this point. However, in the case of this specific system, TIC 120362137, we respectfully disagree. There are codes available which can, in principle, handle both the secular dynamical and stellar evolution models, e.g., <https://ui.adsabs.harvard.edu/abs/2016ComAC...3....6T/abstract>. In this latter code, TrES, the stellar evolution part is handled by the fast (i.e., approximation) stellar evolution code SeBa.

For TIC 120362137 we have (as noted by the Reviewer) separately investigated the dynamics of this highly hierarchical and coplanar system. We find that not much dynamically interesting happens over the shorter term (million years), and that the system is stable for Gyr, implying that also nothing dramatic happens dynamically over the longer term.

We further look at the sequential sets of mass transfer from Aa to Ab, then from A to B, and finally from AB to C. We use MESA to show that the first mass transfer event is dynamically unstable and leads to a merger. This is more robust than using a code like SeBa (or other approximation codes) to determine the mass-transfer stability. There is no reason for a large amount of mass to be lost from this merger, in which case the orbits of B and C are largely unaffected by the merger. Likewise, we find similar results for the merger of A and B, and so forth. MESA is at least as good as, and in most cases better than, any other code designed for understanding what happens during the initial phases of mass transfer, and if a common envelope will ensue.

And, by the way, MESA was set up explicitly from very near its inception to handle the simultaneous evolution of binary stars undergoing mass transfer. It stops when the Roche lobe of the accretor becomes filled, but by that time the die has been cast, and one knows whether the transfer will lead to a common envelope. We know this to be the case since one of the authors on this paper worked with Bill Paxton many years ago to incorporate binary mass transfer into MESA.

We therefore see no better way to evaluate the future evolution of this system than the way we have done it. It is no less certain nor more speculative than anything TrES could do for TIC 120362137, and therefore we hold fast in wanting to keep this part of the paper as it is. Despite this, however, as it will be discussed below, now we decided to remove the MESA evolutionary analysis from the main text, and put it into Supplementary Material.

Regarding the binary evolution, the statements on common envelope evolution and Roche lobe overflow are speculative. The system is likely dynamically stable, and the binary's mass transfer stability should be reassessed. I suggest removing the binary (quadruple) evolution section and Figure 6.

As discussed above, we respectfully disagree with the referee. Despite this, however, in accord with the suggestion of the Reviewer, we removed almost all the evolutionary study, as well as Fig. 6 from the main text, and we mention there (in the main text) only that certain fact, that the primary star (Aa) of the inner EB will fill its Roche-lobe first. We intend to keep the evolution study, as well as the former Fig. 6, as a part of the Supplementary Material. In this latter material, we have, added a couple of prefatory sentences to the future evolution where we comment about the degree of uncertainty inherent in such calculations.

The manuscript remains longer than necessary. Please focus on the observational data and key conclusions, leaving deeper theoretical interpretation to future work. The writing could still be polished to reduce colloquial expressions.

Removing the evolutionary study, together with the corresponding Figure 6, from the main text, and putting them into the Supplementary Material, made the paper naturally shorter. We feel, that this may satisfy this recommendation of the Reviewer.

It is commendable that the number of self-citations has been reduced, reflecting the first author's contributions more fairly. Thank you for addressing this.

To summarize, previous recommendations mostly stand:

Reconsider the evolutionary scenario for the inner binary; current treatment is inadequate and should be removed.

As we discussed above in detail, we respectfully do not agree with the Reviewer that our treatment is inadequate.

Replace the stability analysis based on direct integration with an appropriate secular dynamical analysis or remove it.

We discussed this question in detail above. In the end, we removed the direct integration part of the stability analysis.

Focus the manuscript more tightly on observations, reducing length and speculative content.

The manuscript, in our view does focus on the observations, and is now as short as we can make it without losing important content. In this regard, we note that the entire Section 2 discusses the current system parameters, obtained directly from the observations, and compares this quadruple to other known systems, while Section 4 gives all the observational details, as well as describes the methods which were used for the analysis of the observations. Moreover, Section 1 places this quadruple simply into

context. Only the second half of Sect. 3 was devoted to discussing the potential evolutionary scenarios. Putting this evolutionary scenario into the Supplemental Material, we now feel, that the manuscript itself will be not only shorter, but fulfil even those expectations of the Reviewer, that it remains now more tightly concentrated on observations. Moreover, counting the figures, now there are five observational figures in the main text, that is, all the five illustrate either the observationally well-constrained configuration of the system, or show direct observations together with fitted models. Therefore, we feel, that we have found a balanced ratio between well-constrained facts and “uncertain future predictions”.

Moderate subjective language and emphasize well-supported conclusions.

It is not absolutely clear to us what the Reviewer meant under “subjective” language. We dropped out, however, several expressions which, in our view, might have been categorized as “subjective”. For the convenience of the Reviewer, we did not delete them, but simply strike out these expressions, with the understanding that they will be deleted after the manuscript has been reviewed again.

We also changed the text to the passive voice instead of first person at several places.

Focus discussion on well-constrained system history rather than uncertain future predictions.

If by “well-constrained system history”, the referee means the formation scenario for this type of system, then we respectfully disagree that this is something that can be discussed with any certainty. We do refer to Tokovinin’s review article (Tokovinin, 2021, Universe, 7, 352) for an overview on how such systems may form.

Address the argument for common-envelope evolution in a radially stable non-Darwin unstable system, which remains unaddressed.

We don’t understand what a “radially stable” system is. We have explained clearly why the mass transfer in this inner binary is dynamically unstable. This is something much better assessed by MESA than by using other approximation stellar evolution codes.

As a final remark and recommendation. It might be better to focus on the discovery, and that excitement, rather than putting any emphasis on the dynamical and stellar evolution of the system. It turns out too much of a challenge to say anything that makes sense at this point. Adopt Aarseth-Marding and the system's age to argue it's dynamical stability, and report on the primary being the first to leave the main-sequence and commencing in RLOF. Leave it by that, and do not try to predict the system's future by unreliable stellar evolution calculations or dynamical simulations.

We believe we have covered these points above.

Thank you for your efforts.